# Chromatin fibers stabilize nucleosomes under torsional stress

Artur Kaczmarczyk [1,2,3], He Meng[1], Orkide Ordu[2], John van Noort [1]* & Nynke H. Dekker[2]*

Torsional stress generated during DNA replication and transcription has been suggested to facilitate nucleosome unwrapping and thereby the progression of polymerases. However, the propagation of twist in condensed chromatin remains yet unresolved. Here, we measure how force and torque impact chromatin fibers with a nucleosome repeat length of 167 and 197. We find that both types of fibers fold into a left-handed superhelix that can be stabilized by positive torsion. We observe that the structural changes induced by twist were reversible, indicating that chromatin has a large degree of elasticity. Our direct measurements of torque confirmed the hypothesis of chromatin fibers as a twist buffer. Using a statistical mechanics-based torsional spring model, we extracted values of the chromatin twist modulus and the linking number per stacked nucleosome that were in good agreement with values measured here experimentally. Overall, our findings indicate that the supercoiling generated by DNA-processing enzymes, predicted by the twin-supercoiled domain model, can be largely accommodated by the higher-order structure of chromatin.

[1] Huygens-Kamerlingh Onnes Laboratory, Leiden University, Niels Bohrweg 2, 2333 CA Leiden, The Netherlands. [2] Department of Bionanoscience, Kavli Institute of Nanoscience, Delft University of Technology, van der Maasweg 9, 2629 HZ Delft, The Netherlands. [3] Present address: Faculty of Medicine, Imperial College London, Du Cane Road, W12 0NN London, United Kingdom. *email: noort@physics.leidenuniv.nl; N.H.Dekker@tudelft.nl

In eukaryotic cells, the accessibility of DNA is regulated at the level of chromatin. The fundamental repetitive unit of chromatin is the nucleosome, which consists of 147 base pairs of DNA wrapped around a wedge-shaped histone octamer composed of an $(H3-H4)_2$ tetramer and two H2A-H2B dimers[1,2]. Normally, nucleosomes constitute a barrier for polymerases and other enzymes involved in DNA transactions[3]. However, they are dynamic structures that can be influenced by e.g., thermal fluctuations[4,5], DNA supercoiling[6], post-translational histone modifications[7], activity of chromatin remodellers[8–10], or the presence of linker histones[11,12]. These mechanisms provide the means to modulate nucleosome stability, and hence influence the structure of chromatin.

In the genome, nucleosomes are spaced by so-called linker DNA of 20–90 bps[13,14], but they can nonetheless interact with each other via histone tails to form higher-order chromatin structures, e.g., fibers[15,16]. The topology of such chromatin fibers in vivo has remained a matter of debate, in part due to the heterogeneity of chromatin and its dynamic nature[17–19]. Indeed, the latest improvements in chromosome conformation capture methods and microscopy techniques allowed to reveal various structural motifs between nucleosomes in situ[20–22]. While many detailed models have emerged from structural studies of nucleosomal arrays in vitro and in silico[15,23–25], the question of how the stability of nucleosomes and the accessibility of the DNA can be modulated in compact chromatin remains not explained. Possibly, stacking of nucleosomes into chromatin fibers imposes an additional barrier for molecular motors that act on DNA, and as such contributes to gene regulation.

A primary regulator of gene expression is supercoiling, the over- or under-twisting of the right-handed DNA double helix (relative to its canonical 10.4 base pairs per helical turn[26]) that results from bending or unwinding of DNA during replication or transcription. During the latter process, RNA polymerase (RNAp) generates large torsional stresses at rates up to seven DNA supercoils per second[27]. More specifically, as described by the "twin-supercoiled domain" model, RNAp overwinds DNA ahead of the transcription fork and underwinds DNA behind it[27–29]. It has been suggested that the positive torsion ahead of a eukaryotic RNAp could destabilize nucleosomes[30], whilst the negative torsion in its wake could facilitate their reassembly (Fig. 1a). Such phenomenon was initially observed in vivo by Teves and Henikoff[31], who employed an MNase digestion assay to resolve twist-induced changes in nucleosome occupancy. A genome-wide study of the supercoiling density within large topological domains also showed that overwound and underwound regions differed in the degree of chromatin compaction[32]: actively transcribed, gene-rich loci were typically found to be underwound yet less compacted relative to transcriptionally silent regions of densely packed chromatin. This complex interplay between transcription, supercoiling, and chromatin compaction has been investigated at a mechanistic level only in the context of individual nucleosomes[33]. Thus, it remains unknown how torsional stress affects the compact chromatin fibers formed by multiple interacting nucleosomes.

To understand the propagation of twist in chromatin, it is crucial to resolve the topology of DNA in a folded chromatin fiber. The nucleosomal DNA winds around a histone octamer 1.65 times in a left-handed manner[2] and has a slightly lower helical twist density than bare DNA. As a result, the change of linking number (corresponding to the sum of twist and writhe of DNA) of a single nucleosome equals $-1$, as inferred from direct measurements on DNA minicircles[34]. However, it does not follow that the topological properties of chromatin fibers can simply be inferred from the behavior of individual nucleosomes, because the linker DNA that connects nucleosomes forms additional loops

due to nucleosome stacking[15,35–37]. Thus, an understanding of the impact of twist generated on chromatin fibers by DNA-processing enzymes should take into account the higher-order structure of chromatin.

A detailed view of the dynamics of condensed chromatin fibers and their unfolding has been provided by in vitro single-molecule force- and torque- spectroscopy techniques[36,38–42]. For example, using force-spectroscopy we studied the compliance of condensed fibers assembled with different nucleosome repeat lengths (NRLs) and inferred that chromatin arrays with short linker DNA of 20 bp (NRL = 167 bp) fold into two-start fibers, whereas those with longer linker DNA of 50 bp (NRL = 197 bp) fold into one-start, solenoidal fibers[36,40]. We also characterized the impact of stacking interactions by stretching chromatin fibers with H4-tail-cross-linked nucleosomes[36]. Interestingly, work on isolated fragments of native chromatin showed remarkably similar unfolding behavior; however, large heterogeneities were observed in terms of composition[42]. These experiments, all performed on rotationally unconstrained molecules, collectively indicate that nucleosome–nucleosome interactions form the driving force behind chromatin condensation and contribute to their resilience under mechanical stress.

Single-molecule experiments focusing on the role of torsion have examined mainly individual nucleosomes or sparsely decorated chromatin fibers. Thanks to the high spatial- and temporal resolution of these techniques, it has been also possible to investigate the dynamics at the sub-nucleosomal level. For example, it was demonstrated by Vlijm et al.[43] that tetrasomes (complexes of 80 base pairs of DNA wrapped around a $(H3-H4)_2$ tetramer) spontaneously flip between left- and right-handed wrapped states of the DNA, suggesting that they could contribute to the torsional plasticity of chromatin[44]. Similar transient states were not detected in nucleosomes; yet, isolated nucleosomes can absorb some of the imposed torsional stress by buffering the twist defects in the DNA helix and thereby reduce the build-up of torque[43,45]. Earlier, Sheinin et al.[46] investigated the force-induced unfolding of single nucleosomes from a rotationally constrained DNA template. Here, the application of force to pre-twisted nucleosomes primarily facilitated the eviction of H2A-H2B dimers and had only a moderate effect on the $(H3-H4)_2$ tetrasome unwrapping relative to rotationally relaxed samples. The response to twist of rotationally constrained nucleosomal arrays in a bead-on-the-string configuration was investigated by Bancaud et al.[47,48], who reported a large degree of torsional plasticity of chromatin and attributed this to a chiral transition of nucleosomes from a left-handed into a right-handed wrapped state. However, as the use of a low salt buffer in these experiments prevented the folding of nucleosomal arrays into compact fibers, neither the stacking between nucleosomes nor DNA unwrapping from nucleosomes could be probed. Under conditions that more closely resemble the in vivo context (i.e., 100 mM KCl and 2 mM $MgCl_2$), nucleosomes are known to stack into a condensed fiber[16], and the torsional response is therefore expected to differ.

In this work, we report a comprehensive study of chromatin dynamics in physiological salt concentrations under force and torque. By employing DNA with multiple repeats of 601-sequences, we obtain uniform chromatin fibers stabilized by histone-tail interactions[36]. We find that chromatin fibers respond anisotropically to torsion and, importantly, maintain their compacted higher-order structure. Based on quantitative analysis, we can establish structural parameters of folded chromatin fibers, such as the topology of the linker DNA and the chirality of the fiber. Similar phenomena are detected for chromatin fibers with a nucleosome repeat length = 167 bp and 197 bp. These results indicate that both types of fibers form left-handed superhelices that can absorb excessive positive twist. Such elastic response

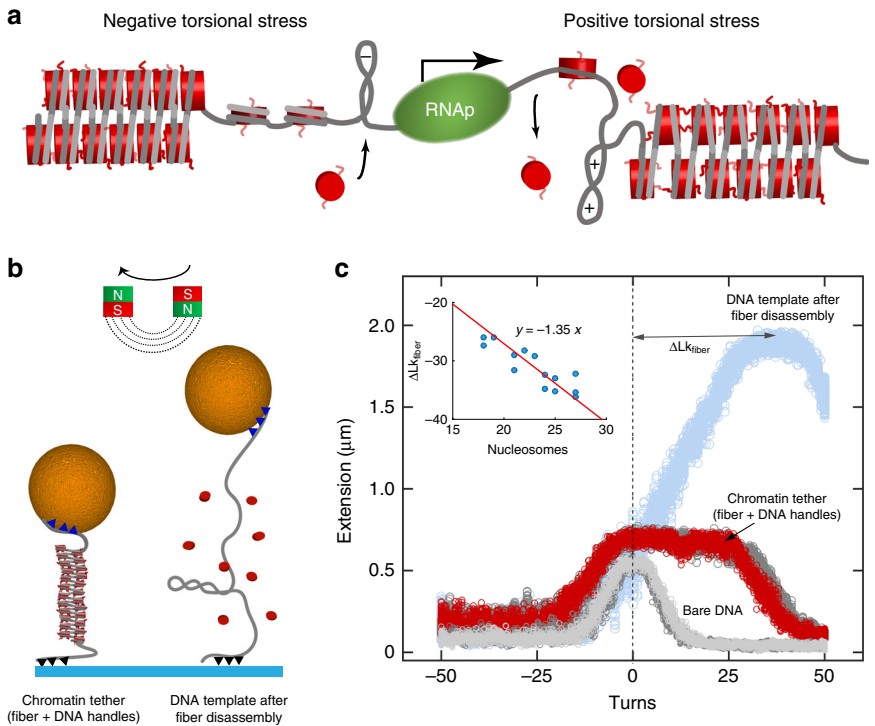

**Fig. 1 Applying turns to reconstituted chromatin fibers mimics the effect of torsional stress in vivo. a** Schematic summarizing the current understanding of the response of chromatin to transcription-generated torsional stress. Progressing RNA polymerase (RNAp) generates positive DNA supercoils, ahead of the transcription bubble, that destabilize nucleosomes (red). Negative supercoils that form behind RNAp facilitate nucleosome assembly. **b** Schematic of single-molecule magnetic tweezers used to apply tension and torsion to reconstituted chromatin. Fibers containing ∼30 regularly spaced nucleosomes (histone octamers, red, wrapped by DNA, grey) are flanked by ∼2 kb DNA handles for tethering to a paramagnetic bead (gold) and a glass slide (blue), respectively. The DNA handles include either multiple biotin or multiple digoxigenin moieties (triangles) in order to ensure rotational constraint. A pair of cubic magnets (red and green squares) exerts a constant stretching force on the paramagnetic bead, and hence on the tether. To apply rotations to the tethers, the magnets are rotated in either a positive (counter-clockwise) or negative (clockwise) direction. **c** Rotation-extension curves of DNA (grey) and a chromatin tether (red) under a force of 0.5 pN. At this force, the rotation curve of DNA handles alone (grey) is symmetric with respect to zero turns. The rotation curve of a chromatin tether (red) is asymmetric with respect to zero turns and exhibits a broad apex at positive turns. Note that this dataset obtained by rotating from negative to positive turns fully overlaps with a subsequent dataset obtained by rotating in the reverse direction (dark grey). Upon the addition of heparin to the same tether, the rotation–extension curve (blue) exhibits a maximal extension that exceeds that of the folded fiber by a factor of three. Furthermore, the apex of the curve is shifted by +34 turns, from which the negative linking number of the folded chromatin fiber is deduced (Lk = −1.4, see main text). Inset: the linking number of a chromatin fiber, quantified by the shift between rotation–extension curves of chromatin fibers versus bare DNA, is proportional to the number of assembled nucleosomes. The red line is a linear fit to the data with a slope of −1.35 (SE = 0.02) that corresponds with the population-averaged linking number per nucleosome ($n = 14$). Underlying rotation–extension curves are shown in Supplementary Fig. 1. Source data are provided as a Source Data file.

allows chromatin to function as a twist reservoir that can accommodate positive supercoiling.

## Results

**Left-handed chromatin fibers absorb positive twist.** We studied torsional properties of chromatin fibers reconstituted on tandem repeats of 601-DNA sequences (30·167 bp and 25·197 bp) flanked by 2030 bp nucleosome-free DNA handles (jointly referred to below as the 'chromatin tether'). Previously, we reported that torsionally relaxed chromatin tethers undergo structural changes by the application of force range of 2–40 pN[36,40,42]. Here, the samples were tethered at either end by multiple bonds to a magnetic bead or to the surface of a flow cell, respectively, in order to constrain their overall linking number (Fig. 1b) and to capture the structural dynamics of chromatin fibers under torsion.

We first performed a series of experiments in which we measured the extension of the 167-nucleosome repeat length (NRL) chromatin tethers at constant tension of 0.5 pN while applying a sequence of 50 positive turns, 100 negative turns and 50 positive turns (thereby returning to the starting point). In the absence of applied turns (designated as zero turns in all plots), the tether was assumed to be in a fully torsionally relaxed state. The resulting rotation–extension diagram of the chromatin tether (Fig. 1c, red) showed an asymmetric response of chromatin to applied positive and negative twist. In particular, the extension exhibited a broad plateau that persisted for more than 25 positive turns. This differs substantially from the response of a bare DNA molecule of comparable length to applied turns (Fig. 1c, grey). Beyond 25 turns, applying more positive turns to the chromatin tether induced a linear shortening (or buckling), as is also observed for a bare DNA once the accumulation of positive twist had led to the build-up of a critical torque[49]. The delay in buckling of the chromatin tether relative to bare DNA tether implies that the chromatin fiber was able to absorb some positive twist, reducing the build-up of torque and thereby delaying the formation of plectonemic supercoils (or plectonemes) in the DNA handles. In contrast, when negative turns were applied, the buckling occurred after applying the same number of negative turns as in bare DNA (−5 turns). This shows that negative twist could not be absorbed by the chromatin fibers

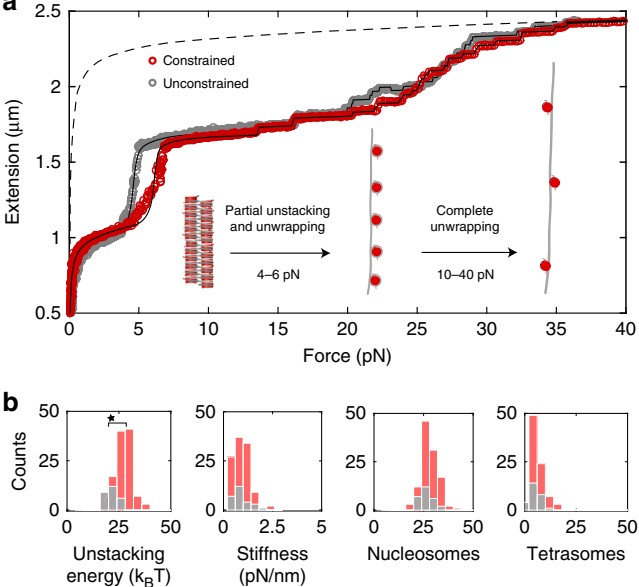

**Fig. 2 The stacking of nucleosomes is stabilized by the presence of rotational constraint. a** Force-extension (F–E) curves of 167-NRL chromatin fibers assembled on a rotationally constrained (red) or unconstrained (grey) 601-DNA template. Both curves were fit to the statistical mechanics model (black lines) derived in our earlier work[40]. From these fits, the number of nucleosomes $N_{nucleosomes}$ and the number of tetrasomes $N_{tetrasomes}$ in the fibers were determined. These equaled 25 ±1 and 1 ± 1, respectively, in both tethers. The F–E response for 7.0 kb DNA according to the WLC model is co-plotted (dashed line). Below 1 pN, the response of the tethers to force reflects the entropic elasticity of the DNA handles that flank the folded chromatin fiber. Subsequently, in a linear regime between 1 and 4 pN, the stretching elasticity of the fiber dominates the F–E curve. Beyond 4 pN, in the rotationally unconstrained tether, the unstacking transition is observed, which results in a gain in the extension of nearly 0.5 μm due to the release of linker DNA between nucleosomes and ∼56 bp of unwrapped DNA per nucleosome. In the rotationally constrained tether, the unstacking transition occurs at a higher force of 6.5 pN. Further increase of force results in a gradual increase in the extension (∼10 bp per nucleosome)[40] and from ∼10 pN onwards, in a step-wise unwrapping of the remaining 80 bp of DNA from each octamer (or tetramer). Beyond 35 pN, the extension of the tether equals that of bare DNA, indicating complete unwrapping of DNA from the nucleosomes. The inset depicts the conformational changes of DNA and chromatin induced by the increasing force. **b** Distributions of parameters that result from quantitative analysis of rotationally constrained ($n = 112$, red) and unconstrained ($n = 27$, grey) 167-NRL fibers with the statistical mechanics model[40]. The fibers contain on average 27 ± 3 and 27 ± 4 (mean ± SD) assembled core particles (either nucleosomes or tetrasomes), respectively. The deduced stretching stiffnesses of 1.0 ± 0.5 (for rotationally constrained fibers) and 0.9 ± 0.4 pN·nm$^{-1}$ (for rotationally unconstrained fibers) are similar to those obtained in our previous work[36,40]. The mean unstacking energy of 27 ± 4 $k_BT$ in rotationally constrained fibers is significantly larger (Student t-test, p-value = 0.05) than in unconstrained fibers (21 ± 2 $k_BT$). Source data are provided as a Source Data file.

induced torques studied here. Jointly, the extensive plateau and lack of hysteresis in the rotation–extension curves indicate that chromatin fibers can accommodate substantial torsional stress while maintaining nucleosome–nucleosome stacking.

We subsequently quantified the amount of twist stored in a relaxed folded chromatin fiber. To do so, the sample described above was exposed in situ to heparin. This polyanion out-competes the electrostatic DNA-histone interactions, resulting in the disassembly of nucleosomes[51]. Dissociation of a single histone octamer from rotationally constrained DNA increases the contour length by > 50 nm but does not change the linking number of the tether, unless the magnetic field is removed or rotated[43,47]. Following the heparin treatment, the maximal extension of the nucleosome-free DNA was indeed ∼3 times larger than that of the folded fiber (Fig. 1c, blue), indicative of nucleosome disassembly. The most extended state, corresponding to rotationally relaxed DNA, was observed at +34 turns. By determining the number of nucleosomes (and tetrasomes; see Supplementary Note 1) for each individual fiber using fits to their respective force-extension curves[40], we could estimate the linking number of a fiber per nucleosome $Lk_{fiber}$. In the chromatin tether shown in Fig. 1c ($N_{nucleosomes} = 25$), this parameter equals $\frac{-34}{25} = -1.4$. A plot of $Lk_{fiber}$ as a function of the number of assembled nucleosomes for 14 different 167-NRL chromatin fibers (see representative traces in Supplementary Fig. 1) is shown in the inset of Fig. 1c, wherein we included a correction of $\Delta Lk = -0.4$ per tetrasome present[43]. The slope of a linear fit to this data yielded a more accurate measure of the $Lk_{fiber}$ of $-1.35 \pm 0.02$. This is a larger negative number, by $-0.35$, than the linking number of a single, isolated nucleosome[34]. We attribute the more negative linking number per nucleosome to the higher-order structure of the chromatin fiber, in which segments of linker DNA are constrained in such a way as to impart negative chirality to the fibers. In summary, chromatin fibers with 167-NRL thus fold into a left-handed superhelix that does not overwind upon the application of negative turns. Nevertheless, they can be underwound by the application of positive turns until the accumulated torque in the chromatin tether exceeds the DNA buckling torque and plectonemes form in the DNA handles.

**Nucleosome unstacking is influenced by torque.** The higher-order structure of chromatin fibers is maintained by the stacking interactions between nucleosomes[36,37]. We compared the forces necessary to rupture these interactions and unwrap DNA from the nucleosomes in rotationally unconstrained chromatin tethers versus rotationally constrained ones (the latter were selected to have nicks in one of the DNA strands, see Methods). The force-induced conformational changes in chromatin fibers with 167-NRL are shown in Fig. 2a. The unfolding profiles below 4 pN are identical both in rotationally unconstrained (grey) and con-strained molecules (red), indicating that the fiber compliances remain the same. The profiles diverge, however, in the range of 4–6 pN. Here, the abrupt increase in the extension (by nearly 0.5 μm) is associated with the unstacking transition, which comprises unstacking of nucleosomes, release of linker DNA and unwrapping of 56 bp of DNA per nucleosome[40]. Notably, a higher force (by nearly 2 pN) was required to rupture nucleosome–nucleosome interactions in the rotationally con-strained tether relative to the unconstrained one. At higher forces between 13 and 35 pN, the remaining wrapped DNA was released in discrete steps in a manner that was independent of rotational constraint.

Fitting the force-extension curves of rotationally constrained and unconstrained 167-NRL fibers to a previously developed statistical mechanics model[40] yielded the number of nucleosomes

and instead propagated to the DNA handles, resulting in buckling once a critical torque had built up[49,50]. Importantly, all rotation–extension curves were fully reversible (red and dark grey curves in Fig. 1c), as opposed to the magnetic tweezers experiments on sparsely decorated nucleosomal arrays[48]. This indicates that the structural changes occurred in thermodynamic equilibrium and that histone disassembly can be excluded at the applied forces and

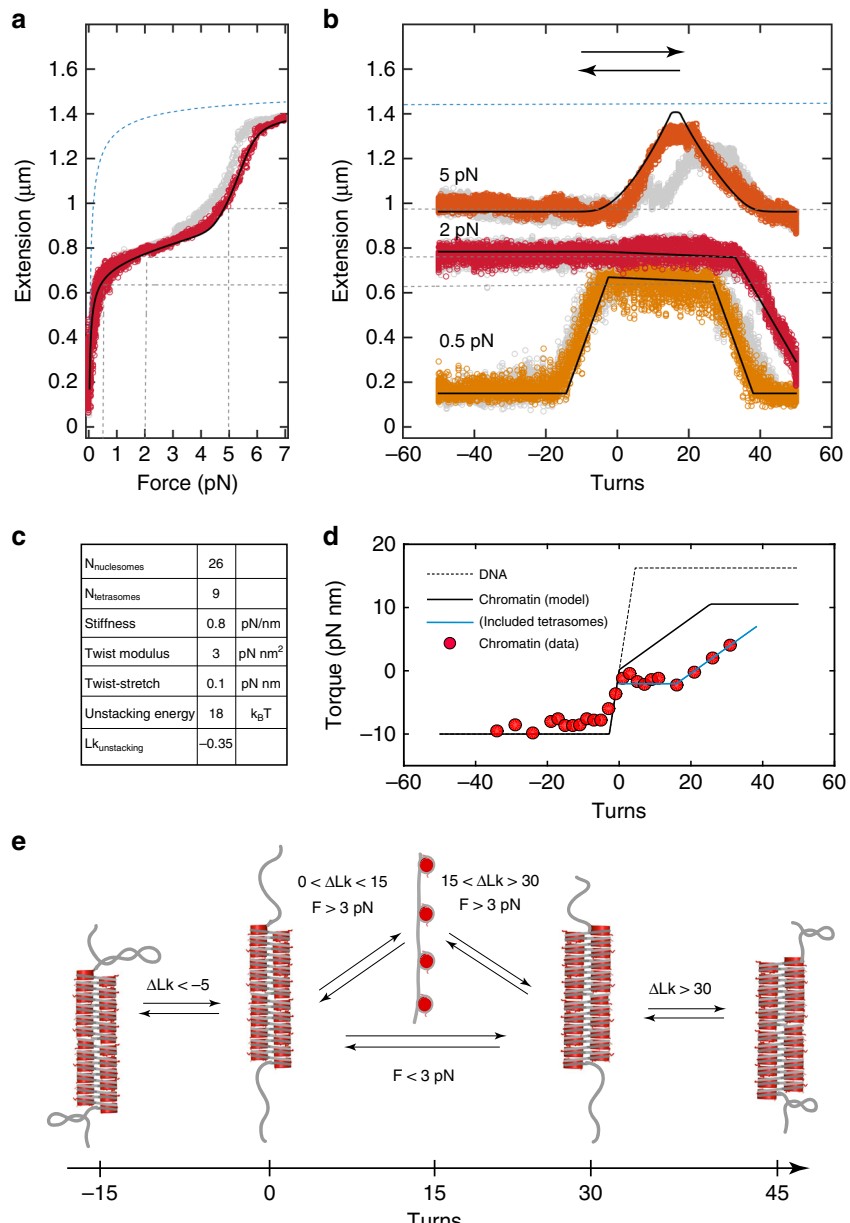

**Fig. 3 167-NRL fibers adopt a left-handed, two-start structure that refolds upon excessive positive turns. a** Force-extension curve of a rotationally constrained 30·167 chromatin fiber. The initially compacted fiber is stretched under force (red) and observed to unstack at 5.5 pN. When the force is subsequently decreased, the fiber is observed to refold to its initial configuration (grey). The solid black line reflects the best fit to the statistical mechanics model with $N_{\text{nucleosomes}} = 26$, $N_{\text{tetrasomes}} = 9$, unstacking energy $G_u = 18\ k_B T$ and stretching stiffness per nucleosome $c_s = 0.7\ \text{pN·nm}^{-1}$, in line with the results shown in Fig. 2B. The dashed blue line reflects the extension of a DNA template with singly wrapped nucleosomes. The dashed black lines mark the forces and extensions examined during the subsequent rotation experiment. **b** Rotation-extension curves of the same molecule as in panel **a** at forces of 0.5 pN (orange), 2 pN (dark red) and 5 pN (light red). Grey curves reflect the measurements with a reversed direction of magnet rotation. Solid black lines reflect the torsional spring model using the parameters as in panel **a** together with a torsional stiffness $c_t = 3 \pm 1\ \text{pN·nm}^2$ (mean + SD) and a twist-stretch coupling factor $c_{ts} = 0.1\ \text{pN·nm}$. **c** Table summarizing the parameters used in the quantification of curves in panels **a** and **b**. **d** Representative torque-twist measurement on a 167-NRL chromatin fiber at 2 pN (red). Data was offset to equate the torque plateau at negative twist with the DNA melting torque of -10 pN·nm. The dotted grey line represents the predicted torque build-up in the DNA handles alone. The solid black line shows the restoring torque of the chromatin tether quantified with a torsional spring model yielding a torsional stiffness $c_t = 3\ \text{pN·nm}^2$. The solid blue line is the prognosed torque that includes a correction for twist in tetrasomes. **e** Schematic summarizing the structural changes in the chromatin tether under torsion. Torsionally relaxed 167-NRL fiber (twist = 0) cannot absorb negative turns, and as such the entire linking number change is absorbed by the DNA handles. The consequences of imposed positive twist depend on the level of tension and may involve unstacking of chromatin fiber followed by changes in its chirality. This delays the formation of DNA plectonemes compared to negative twist.

in each chromatin fiber (on average 25), as well as a number of physical parameters that characterize the individual fiber. For example, for rotationally constrained tethers, we obtained an unstacking free energy of 27 ± 4 $k_BT$ (mean ± SD, $n = 112$), compared to 21 ± 2 $k_BT$ for unconstrained tethers ($n = 27$) (Fig. 2b). In the low force regime, the stretching stiffness of the chromatin fibers equaled 1 ± 0.5 pN·nm$^{-1}$ in the presence of rotational constraint, and 0.9 ± 0.4 pN·nm$^{-1}$ without. In this regime, the extension of 167-NRL fibers is unaffected by the presence of rotational constraint. For 197-NRL fibers, we report similar findings compared to 167-NRL fibers. For example, attaining the unstacking transition required a higher force by nearly 2 pN relative to rotationally unconstrained fibers (Supplementary Fig. 2A). The stretching stiffness (0.2 ± 0.1 pN·nm$^{-1}$), previously measured on rotationally unconstrained fibers to be smaller than that of 167-NRL fibers due to the different geometry of their higher-order structure[36,40], is found to be independent of torsional constraint (Supplementary Fig. 2B).

Overall, the primary consequence of torsional constraint in the context of stretching was the higher force necessary to induce chromatin fiber unstacking. This can be understood by considering that the rupture of internucleosomal contacts in a rotationally constrained folded chromatin fiber must be accompanied by a large decrease in its writhe. At fixed linking number, this must be compensated by increased twist in the DNA handles. It thus appears that the high energy cost of over-twisting DNA (included in the fitted unstacking free energy) will delay the onset of the unstacking transition but does not otherwise alter the unwrapping of the last turn of DNA from nucleosomes (see Supplementary Note 2 for further details).

**Positive twist can stabilize and destabilize chromatin fibers.** We have further characterized the response of chromatin fibers to torsion by twisting fibers at three different forces. Prior to twisting, we performed a stretching experiment, as described above, to assess the fiber composition. For example, by fitting the force-extension curve of the 167-NRL fiber in Fig. 3a, we inferred that it contained more than 30 assembled particles: 25 fully folded nucleosomes, and 9 tetrasomes that result from partial disassembly of H2A-H2B dimers (e.g., from excessive nucleosomes formed on the non-601 DNA handles). As evidenced by presence of the unstacking transition at ~5 pN, the chromatin fiber remained fully folded below this force, allowing us to probe its response to turns. The application of turns at the lowest force of 0.5 pN (Fig. 3b, orange) resulted in the asymmetric rotation–extension curve comparable to the one presented in Fig. 1c (red). The same phenomenon was observed when positive turns were applied at 2 pN (Fig. 3b, dark red). Also here, the extension remained constant up to 25 positive turns, suggesting that the induced twist was absorbed by the chromatin fiber. Only when a larger number of turns was applied, i.e., $\Delta Lk > 25$, did the tether extension shorten, suggestive of the formation of DNA plectonemes in the DNA handles. The application of negative turns at a force of 2 pN did not affect the tether extension. Assuming that the chromatin fiber also does not accommodate negative twist at this force, this would be consistent with the well-established torque-induced melting (or denaturation) of the DNA helix in the flanking handles[49].

To obtain more insight into the origin of the torsional plasticity of chromatin fibers, we used magnetic torque tweezers[52] and performed direct torque measurements (plotted versus the number of applied turns) on the chromatin tether at the same force of 2 pN (Fig. 3d). For comparison, the anticipated build-up of torque in a bare DNA tether is shown

as a dotted line (separately measured data presented in Supplementary Fig. 6C). In short, for bare DNA the application of either positive or negative turns resulted in the build-up of twist in the molecule and a concomitant increase in torque. At negative turns, once the torque equaled the denaturation torque, it saturated and remained constant. At positive turns, once the torque equaled the (force-dependent) critical torque for buckling, it also saturated. For the chromatin tether, we observed that at positive turns the build-up of torque proceeded more slowly compared to bare DNA. Even after the application of 30 turns, the chromatin tether had accumulated less than 10 pN·nm of torque, whereas at the same force, the application of only seven turns to bare DNA would yield a torque of 17 pN·nm. Frequently, the build-up of torque in the chromatin tether was preceded by a near-zero-torque plateau, that could last up to ~20 positive turns (see Discussion). In contrast, the build-up of torque at negative turns closely followed that of the torque curve for bare DNA. This indicates, as did the rotation–extension experiments at the same force, that the application of negative turns to chromatin tethers translates into the build-up of torque in the DNA handles.

When we continued to probe the response of chromatin fibers to the rotation, but now under an applied force of 5 pN, different signatures were observed (Fig. 3b, light red). Upon applying positive turns, the tether extension was found to increase, until at 18 turns a maximal length was reached. This length corresponded to that of a fully unstacked fiber (compare maximal lengths in Fig. 3a, b). Subsequently, further application of turns resulted in shortening of the tether. We attribute this phenomenon to the restacking of nucleosomes into a folded fiber. Thus, the unstacking transition of chromatin is facilitated by the application of positive turns, but excessive positive turns restabilize nucleosome stacking and the reassembly of chromatin higher-order structure, most likely with an opposite handedness.

The folding of the chromatin fiber at excessive twist was confirmed in a separate experiment in which the fiber was pre-twisted in the positive direction prior to a pulling and release cycle (Supplementary Fig. 3A). Under these conditions, the unstacking transition occurred at lower force at $\Delta Lk = +10$ than at $\Delta Lk = 0$ (consistent with a reduction in the nucleosome unstacking force), whereas at $\Delta Lk = +20$ turns the unstacking force increased again (consistent with a re-introduction of nucleosome stacking). In contrast, the unstacking force was not affected by the application of negative turns (Supplementary Fig. 3B). These two experiments, performed either at constant force or at constant twist, revealed an interesting property of chromatin fibers: initially, the application of positive twist decreased the extent to which fibers are folded, but the application of a larger number of positive turns, i.e., >1 turn per nucleosome, refolded the fiber into a condensed state with newly stacked nucleosomes.

**Determination of the twist modulus of chromatin fiber.** To quantify the observed response of chromatin tethers to imposed turns, we analyzed the rotation–extension curves by incorporating a torsional spring model into a statistical mechanics framework[40]. We modelled the two components of each tether (i.e., the chromatin fiber and the DNA handles) as springs with differing torsional properties. As the elastic moduli of bare DNA are known[49,53,54], this allowed us to quantify key elastic properties of the folded chromatin fiber, such as the twist modulus per nucleosome $c_t$ and its twist-stretch coupling factor $c_{ts}$. Furthermore, we could capture the phenomenon of twist-induced chromatin unfolding and measure the associated changes in linking number.

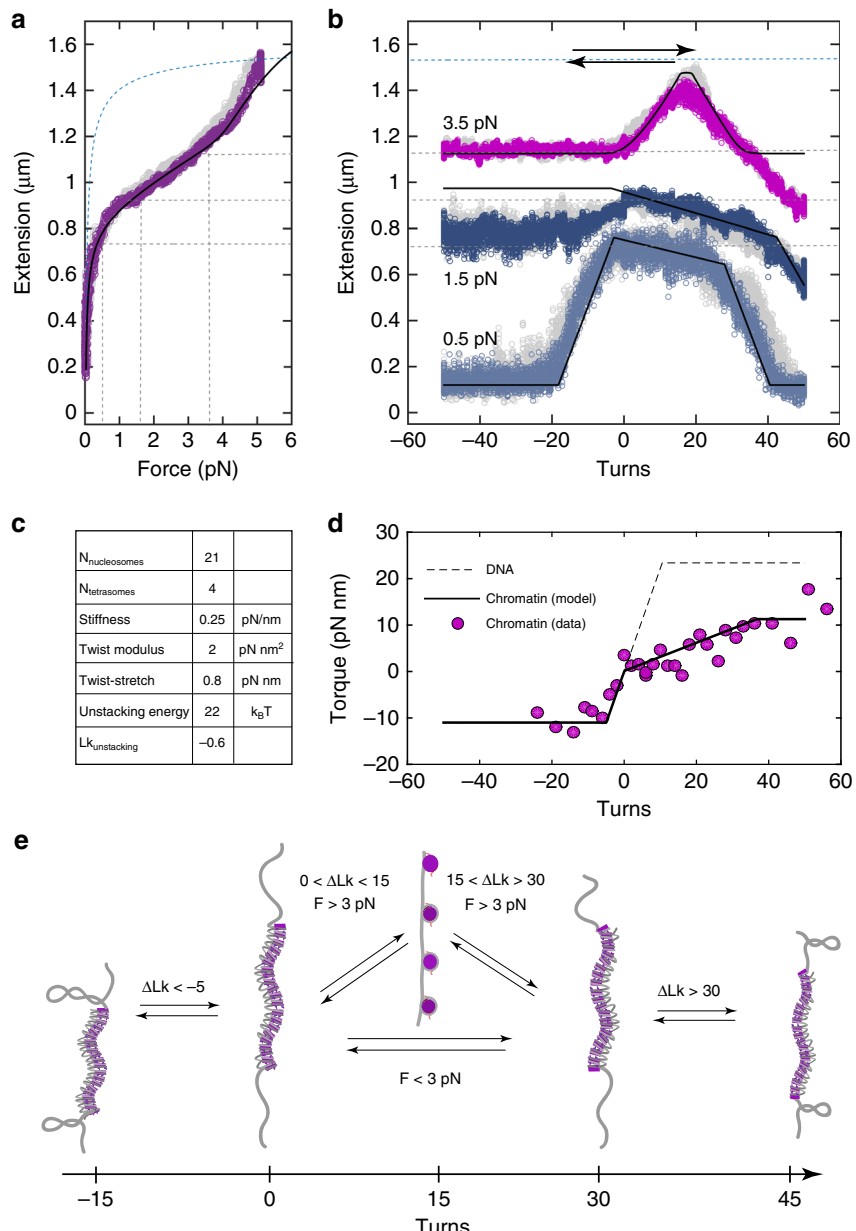

**Fig. 4 197-NRL fibers adopt a left-handed, one-start structure with similar torsional response. a** Force-extension curve of a rotationally constrained 25·197 chromatin fiber. The initially compacted fiber is stretched under force (violet) and observed to unstack at 4.5 pN. When the force is subsequently decreased, the fiber is observed to refold to its initial configuration (grey). The solid black line reflects the best fit to the statistical mechanics model with: $N_{nucleosomes} = 21$, $N_{tetrasomes} = 4$, unstacking energy $G_u = 22$ $k_B T$ and stretching stiffness per nucleosome $c_s = 0.25$ pN·nm$^{-1}$. The dashed blue line reflects the extension of a DNA template with singly wrapped nucleosomes. The dashed black lines mark the forces and extensions examined during the subsequent rotation experiment. **b** Rotation-extension curves of the same molecule as in panel **a** at forces of 0.5 pN (blue), 1.5 pN (dark grey) and 3.5 pN (pink). Grey curves reflect the measurements with a reversed direction of magnet rotation. Solid black lines reflect the torsional spring model using the parameters as in panel **a** together with a torsional stiffness $c_t = 2 \pm 1$ pN·nm$^2$ and twist-stretch coupling factor $c_{ts} = 0.8$ pN·nm. **c** Table summarizing the parameters used in the quantification of curves in panels **a** and **b**. **d** Representative torque-twist measurement on a 197-NRL chromatin fiber at 2 pN (violet). Data were offset to equate the torque plateau at negative twist with the DNA melting torque of -10 pN·nm. The dotted grey line represents the predicted torque build-up in the DNA handles alone. The solid black line shows the restoring torque of the chromatin tether quantified with a torsional spring model yielding a torsional stiffness $c_t = 2$ pN·nm$^2$. **e** Schematic summarizing the structural changes in a chromatin tether under torsion. Torsionally relaxed 197-NRL fiber (twist = 0) cannot absorb negative turns, and as such the entire linking number change is absorbed by the DNA handles. The consequences of imposed positive twist depend on the level of tension and may involve unstacking of nucleosomes in the chromatin fiber followed by changes in its chirality. This delays the formation of DNA plectonemes compared to negative twist.

The input parameters of the model that are fixed for all tethers include, for the DNA, the stretching persistence length $P$, the intrinsic (force-independent) torsional persistence length $C_{lim}$, and the persistence length of a plectonemic DNA $C_p$; and for the chromatin fibers, the contour length per nucleosome for each of the unfolding intermediates: $L_{unstack}$, $L_{unwrap}$[54]. Tether-specific input parameters, assessed from the corresponding force-extension curves, include the contour length of the DNA handles

$C_L$, the number of nucleosomes $N$, and the number of tetrasomes $T$, the stretching stiffness per nucleosome $c_s$, and the free energy of the unstacking transition $G_u$ (see the theoretical framework of the model in Supplementary Note 3).

Validation of the model for DNA alone was performed by reproducing the mechanical response of DNA to applied turns, measured on a molecule consisting of DNA handles only ($N = 0$, $C_L = 2.0$ kbp). We captured the underwinding, overwinding and buckling transitions of DNA using fixed values of $L = 646$ nm, $P = 50$ nm, $C = 100$ nm, and $C_p = 24$ nm. These values were in agreement with previous observations[49]. The deduced values of the effective (force-dependent) torsional persistence length were further validated by an independent measurement of the effective torsional persistence length of DNA in the corresponding buffer conditions using magnetic torque tweezers (Supplementary Fig. 5). Fits of the linear regime yielded a torsional persistence length of 79 nm (SE = 5 nm) at 1.7 ± 0.3 pN (mean ± SD) (Supplementary Fig. 6C, see Supplementary Fig. 7 for torque data with corresponding force-extension curves of DNA), in good agreement with the value that best modelled the rotation–extension curves (Supplementary Fig. 6B). These parameters describing bare DNA were then used in the modeling of the torsional properties of chromatin fibers.

In the torsional spring model, an equal torque $\mathcal{T}$ acts both on the chromatin fiber and the DNA handles. As a result, the applied twist ($\Delta$Lk) is distributed between these two components of our tethers in a manner governed by their torsional properties (see Supplementary Note 3 and Supplementary Fig. 8 for further details). Depending on the twist modulus of the chromatin fiber $c_t$, the applied turns could be differently distributed between the fiber ($\Delta$Lk$_{fiber}$) and the DNA handles ($\Delta$Lk$_{DNA}$). We obtained the best match with the data in Fig. 3b using the following force-dependent values of the effective twist modulus: $c_t = 2$ pN · nm$^2$ at 0.5 pN and 3 pN · nm$^2$ at 2 pN and 4 pN · nm$^2$ at 5 pN. As shown in Supplementary Fig. 8, smaller (larger) values of the twist modulus $c_t$ delayed (expedited) the appearance of the buckling transition in the DNA handles. When divided by the value of the thermal energy $k_B T$ (4.1 pN · nm at room temperature), this yields a torsional persistence length of ~1 nm, which is hundred times lower than that of DNA. The values of the twist modulus were obtained by modelling experimental rotation–extension curves at forces where nucleosomes remained stacked, indicating that chromatin fibers have high torsional flexibility.

Lastly, to describe the unstacking and restacking observed in the rotation–extension curve at 5 pN, we included the chromatin handedness (deduced from the asymmetry of the rotation curve in Fig. 3b) in the energy term of the torsional Hookean spring (see Supplementary Eq. 11). By setting the unstacking energy $G_u$ to 18 $k_B T$, a value close to that measured for rotationally unconstrained fibers, we deduced the linking number change per the unstacked pair of nucleosomes. The modelled curve at 5 pN (Fig. 3b, light red) accurately captures the unstacking and restacking transition, including the characteristic peak in the curve, when the value of $\Delta$Lk$_{stack}$ is set to $-0.4$. Notably, this value is consistent with that deduced from the heparin experiments described above (see Discussion and the overview of parameters of ten independent 167-NRL fibers in Supplementary Table 1). Overall, from applying the torsional spring model to the data, we observed that all fibers display a high torsional flexibility and a negative linking number per stacked nucleosome.

**Torsional properties of chromatin fibers do not vary with NRL**. We have also examined several aspects of the torsional response of 197-NRL fibers. As in the case of 167-NRL fibers, we studied the force-induced unfolding of rotationally constrained fibers both in a relaxed state (Supplementary Fig. 2) and in the presence of supercoiling (Supplementary Fig. 4). We also measured the extensions and torques in these fibers at constant force and under excessive linking number change (Fig. 4b, d). As for 167-NRL fibers, we applied a torsional spring model to the data and deduced whether the distinct fiber geometry, resulting from the longer nucleosome repeat length, resulted in altered torsional properties.

In the stretching experiments, we observed that the stiffness of rotationally constrained 197-NRL fibers (0.2 ± 0.1 pN·nm$^{-1}$, mean ± SD) was the same as in the rotationally unconstrained tethers (Supplementary Fig. 2B). More importantly, these values were on average three-fold smaller than in 167-NRL chromatin fibers. In earlier work, we associated the lower compliance with a one-start, solenoidal helix which has a lower stiffness per nucleosome than a two-start, zig-zag fiber[36,40]. We note that recent cryo-EM studies have demonstrated zig-zag folding for fibers with intermediate NRL of 177 bp or 187 bp[25], which were corroborated by force-spectroscopy experiments by Li et al.[55]. For 197-NRL fibers, however, a high-resolution structure has not been so far reported. Earlier electron microscopy on chromatin fibers inferred an interdigitated solenoidal structure for fibers with such NRL, based on the relationship between NRL and fiber diameter[56]. Our force-spectroscopy data obtained on 197-NRL chromatin are most compatible with this interpretation. We note, that chromatin structure critically depends on other factors e.g., specific buffer conditions or the presence of linker histones (H1/H5)[11,57], which could induce alternative fiber topologies.

In our buffer conditions, the application of turns impacted 197-NRL fibers and 167-NRL fibers in similar fashion, whereas their response to force differed in a manner that appeared to derive from differences in their structures. Figure 4b shows that the 197-NRL fiber also absorbed positive twist and exhibited the properties of a left-handed superhelix with a twist modulus of $c_t = 1$ pN · nm$^2$ at 0.5 pN, $c_t = 2$ pN · nm$^2$ at 1.5 pN, and $c_t = 3$ pN · nm$^2$ at 3.5 pN, which makes them slightly more compliant than 167-NRL fibers. We observed, however, that the tether length slowly decreased when positive twist was applied at low tension. We were able to capture this behavior by invoking a twist-stretch coupling factor $c_{ts}$ of 0.8 pN · nm. Thus, quantitative analysis of 197-NRL fibers indicates that their torsional flexibility exceeds that of 167-NRL fibers. This reduced torsional stiffness was supported by direct torque measurements, which exhibited a very shallow slope in the positive torque regime (Fig. 4d). Overall, despite differences in higher-order structures, the values of $c_t$ differ only slightly between 167-NRL and 197-NRL fibers. As a result, the large compliancy of chromatin fiber to twist appears to be universal, irrespective of its higher-order structure.

## Discussion

The supercoiling of DNA is a fundamental factor in regulating gene expression and affecting chromatin structure[58], even in the presence of topoisomerases that neutralize some of the torsional stress in vivo[59–61]. Here, we have employed single-molecule studies to present a comprehensive characterization of compact chromatin fibers under tension and torsion. Our study of rotationally constrained chromatin fibers goes beyond previous single-molecule assays that focused on the dynamics of non-interacting nucleosomes[43,47,48]. In this report, we have extensively probed the mechanical aspects of higher-order structures of chromatin maintained by nucleosome–nucleosome stacking.

In contrast to these previous studies, our force-extension and rotation–extension measurements at forces below 4 pN exhibited no hysteresis between forward and backward stretching or twisting cycles. In particular, we observed the elastic, anisotropic

response under rotational constraint of two types of fibers, 167- and 197-NRL, which were shown earlier to fold into two-start and one-start conformations, respectively[36,40]. Moreover, we inferred that both fibers likely adopt a left-handed chirality and demonstrated their ability to absorb positive twist. In this way, the nucleosomes in chromatin fibers are protected from unstacking at forces below 5 pN and torques below 10 pN·nm. A remarkable observation is that continued application of positive twist resulted in twist-induced unstacking of chromatin, followed by refolding of the chromatin fiber. Overall, we find that the structure adopted by stacked nucleosomes allows for the effective absorption of twist. This implies that chromatin fibers constitute a topological buffer that can absorb some tension and torsion without unfolding of the nucleosomal DNA. Such behavior, observed here on well-defined 601-nucleosomal arrays folded as a one-start or a two-start fiber, is expected to play a role in other fiber-like motifs, e.g., heterogenous clutches of heterochromatin[20] or well-oriented clusters of nucleosomes[22]. We therefore propose that the secondary structure of chromatin can thus influence the in vivo dynamics of DNA-processing by enzymatic motors that induce its supercoiling, and thereby plays an important role in controlling gene expression.

To ascertain that our samples folded into compact chromatin fibers, we carefully evaluated the composition of individual molecules (see Supplementary Note 1). After verifying the nucleosome and tetrasome occupancy, we could quantify the linking number per nucleosome in a folded fiber as: $Lk_{fiber} = \Delta Lk_{nuc} + \Delta Lk_{stack}$. Direct measurement of the linking number change upon assembly of individual nucleosomes in freely-orbiting magnetic tweezers[62] revealed $\Delta Lk_{nuc} = -1.2 \pm 0.3$ turns[43]. Bancaud et al.[47] measured the change in linking number upon histone disassembly from an array of non-interacting nucleosomes and reported $\Delta Lk_{nuc} = -0.8 \pm 0.1$ turns. In our single-molecule assay, in which we ensured via the use of appropriate salt concentrations that nucleosomes were properly stacked into a folded fiber, we directly measured $Lk_{fiber}$, which we found to equal $-1.35 \pm 0.02$ (Supplementary Fig. 1) in 167-NRL fibers. We attribute this extra negative twist, relative to $\Delta Lk_{nuc}$, to the additional looping in the folded fiber that results in superhelix formation. In a 167-NRL fiber, which folds into a left-handed zig-zag superhelix, the linker DNA crosses back and forth between the two stacks of nucleosomes[15]. This implies that a full turn of the linker DNA is constrained by a pair of nucleosomes, from which one would predict $\Delta Lk_{stack} = \frac{-1}{2} = -0.5$. By subtracting the established $\Delta Lk_{nuc} = -1$ from the obtained $Lk_{fiber} = -1.35$ (inset Fig. 1c), we measured the linking number change of the linker DNA as $\Delta Lk_{stack} = -0.35$, which is close to the predicted value, and the difference could be accommodated by under-twisting the linker DNA. Overall, this interpretation is compatible with a superhelical folding of nucleosomal arrays into a zig-zag fiber mediated by the H4-tail internucleosomal contacts. Such bridging, confirmed in our previous single-molecule experiments with H4-tails cross-linked to the acidic patch of H2A[36], stabilizes the looping of the DNA that we showed here quantitatively.

In parallel, we analyzed the linking number $Lk_{fiber}$ of 167-NRL fibers by the modelled and experimentally obtained rotation–extension curves. By using a torsional spring model, we inferred a very similar value of $\Delta Lk_{stack} \cong -0.4$ (Supplementary Table 1). As explained in Supplementary Fig. 8, this parameter was confirmed by analyzing the rotation–extension curves at forces above 3 pN. The largest extensions observed in these measurements (Fig. 3b) equalled those measured in force-extension curves at 7 pN (Fig. 3a), which is the regime of an unstacked array of singly wrapped octamers or tetramers. Knowing the composition of the fiber (Fig. 3c), it follows that 18 positive turns were necessary to induce the unstacking of the 167-

NRL chromatin with 26 nucleosomes (Fig. 3b). This results in an estimation of $\Delta Lk_{stack} = \frac{18}{-26} = -0.7$. This value is larger than those inferred from a torsional spring model or from the shift in the rotation curve, presumably because the twist absorbed by the DNA handles is not taken into account in this simple estimation. Nevertheless, our three different estimations all yield a negative linking number of the looped linker DNA, which indirectly supports previous reports of left-handed chromatin fibers visualized by EM[25,56,63].

Further analysis of the rotation–extension curves at tension above 3 pN reveals a remarkable phenomenon of a twist-induced refolding of the unstacked chromatin fiber. We observed in two separate assays that excessive positive twist refolds both the two-start fiber (Fig. 3b and Supplementary Fig. 3) and the one-start fiber (Fig. 4b and Supplementary Fig. 4). Thus, at sufficient force (>3 pN), the unstacking transition appears to be reversed by torque in an unprecedented manner. The structure of such over-twisted fibers may involve the so-called reversomes (proposed right-handed nucleosomes[64]) or even the reconfiguration of the entire left-handed chromatin fiber to a superhelix with opposite handedness. Importantly, the tensions and torques involved in this phenomenon are appropriate to the in vivo context, as they are within the range accessible to, for example, RNA polymerases[29]. Possibly, such twist-induced refolding could be a functional event during transcription termination; however, further studies are necessary to unravel more details of this striking unstacking-restacking transition.

We also quantified the elastic response of chromatin at the low force regime, in which stacked nucleosomes maintain the higher-order structure. The measured torsional stiffness of folded chromatin fibers appears to be dramatically smaller than that of bare DNA. The torsional modulus per nucleosome obtained by the model translates to a torsional persistence length of ~1 nm, which is about hundred times shorter than that of the DNA. Such torsional plasticity means that chromatin can absorb large twists without building up much torque. This was supported by direct torque measurements in which positive twist applied to the chromatin tether resulted in a smaller build-up of torque than in bare DNA. A maximal torque of ~10 pN·nm was reached at ~+25 turns, after which supercoiling of the DNA handles ensued, as evident from the reduction of the tether length. The torque threshold is lower than that generated by polymerases (~15 pN·nm)[29], suggesting that torsional stress alone cannot induce chromatin unfolding. The slow build-up of torque in chromatin was, in some fibers, enhanced by the presence of a near-zero-torque plateau in the first positive 10–15 turns. This could not be attributed to the disassembly of histones since no simultaneous change in the extension was observed (Supplementary Figs. 9 and 10). We interpret this plateau regime as a transition of free tetrasomes (as shown by Vlijm et al.[43]) that reside on the DNA handles of our reconstituted fibers. We previously proposed that tetrasomes can by themselves constitute a torque reservoir[44], and it appears that this effect matters also in the presence of folded chromatin fibers. Upon correction of the torsional spring model with tetrasomes, the modelled torque curve (Fig. 3d, blue) well aligns with the experimental data and reinforces the goodness of our torsional spring model.

Lastly, we point out how the free tetrasomes could affect the dynamics of plectoneme formation in the handles of chromatin tethers. To properly describe the slope in the rotation–extension curves of chromatin tethers at large twist (Figs. 3b and 4b), however, we had to decrease the torsional persistence length of plectonemic DNA to $C_p = 12$ nm, suggesting a reduction in the diameter of the supercoils. We attribute this decreased size to the presence of tetrasomes on the DNA handles: wrapping DNA

around a tetrasome would reduce the plectoneme loop size accordingly. It should also reduce the buckling torque, in agreement with the observed buckling transition in the rotation–extension curves of chromatin tethers. Indeed, the torque never reached the buckling torque for bare DNA, i.e., ∼20 pN·nm at 2 pN, as most of the twist was absorbed by the chromatin fiber. Thus, a full description of the behavior of folded chromatin requires taking into account the effect of tetrasomes in close proximity to nucleosomes.

In this work, we have tested the ramification of the "twin-supercoiled domain" model for higher-order chromatin structure, and found that chromatin fibers may locally suppress its effects. We consider in this light the interplay between a folded chromatin fiber and a less compact array of non-interacting nucleosomes located near the actively transcribed locus (Fig. 1a). In this situation, the chromatin fiber could absorb the incoming positive supercoiling and prevent the global build-up of torque. The absorption of positive supercoiling by the soft chromatin fiber enables the progression of transcription along the open chromatin in close proximity to RNAp, where locally the force and torque may be sufficient to unstack nucleosomes. The remarkable refolding transition observed at excessive positive torsion may be a self-regulatory response that could function as transcriptional repressor for nearby genes.

Although nucleosome mapping studies showed that nucleosomes can be evenly distributed in vivo[65,66], they are likely to be less regular than the 601-arrays used in our study. Regular spacing may be reinforced by variety of factors such as linker histones or chromatin remodelers[11,14]. These epigenetic regulators can potentially control the formation of condensed structural units that act as twist buffers, when necessary. DNA sequence is another important factor in nucleosome positioning and may play a role in higher-order folding[67]. The synthetic 601-sequence was used to impose controlled positioning, but how it affects the mechanical properties of folded fibers requires further study. As argued here and in previous studies[36,54,68], 167-NRL and 197-NRL fibers fold into different higher-order structures. Nevertheless, both were shown to have a high capacity to absorb twist, suggesting that this may be a more universal feature of folded chromatin fibers.

Based on these findings we propose that the response of chromatin to torsion demonstrated here can be highly relevant at the level of so-called topologically-associated domains (TADs) found in vivo[69], which could isolate such torsional buffers in the genome. These structures have been reported as long-distance transductors of torsional stress that was not released by topoisomerases. Overall, our study indicates that nucleosome stacking significantly impacts the response of chromatin to twist and increases its flexibility. Collectively, the torsional elasticity of the fully folded fibers, in addition to the conformational changes of the tetrasomes, indicate that large amount of twist can be absorbed via structural changes within folded chromatin fibers. All together ensures that torsionally rigid genomic DNA can be efficiently stored in the nucleus and simultaneously processed during its replication or transcription.

## Methods

**DNA constructs and chromatin assembly**. Two pUC18 plasmids (Novagen) with multiple tandem repeats of the Widom 601-sequences were used as DNA templates for chromatin reconstitution. Plasmid 1 contained 30 repeats of the Widom 601-sequences spaced with 20 bp linkers (NRL = 167 bp), whereas plasmid 2 contained 25 repeats of the Widom 601-sequences with 50 bp linkers (NRL = 197 bp). Both plasmids were digested with BsaI and BseYI (New England Biolabs), and the resulting linearized fragments were separated by gel electrophoresis. The bands representing 7040 bp (plasmid 1) and 6955 bp (plasmid 2) DNA (both including 2030 bp of non-601 handles) were extracted and purified by Wizard SV Gel and PCR cleanup kit (Promega). In parallel, 600 bp DNA fragments were prepared with

PCR on the pUC18 plasmid template using 10% biotin-16-dUTP or digoxigenin-11-dUTP. After amplification, these fragments were digested with BsaI and BseYI, respectively, and ligated to the previously linearized 601-DNA[70].

Wild-type human histone octamers (Epicypher) were mixed with the biotinylated 601-DNA arrays and reconstituted into chromatin fibers using salt dialysis. Multiple titrations with increasing histone:DNA ratios were incubated at 4°C in a high salt buffer (2 M NaCl, 1 × TE, final volume of 50 μl) and transferred into mini-dialysis tubes (10000 MWCO, Thermo Scientific). The samples were dialyzed overnight at 4°C against 200 ml of a buffer with gradually decreasing ionic strength down to 100 mM NaCl. The salt gradient was maintained by a peristaltic pump (Bio-Rad Econo Gradient Pump, flow rate 0.9 ml·min$^{-1}$)[70,71]. Quality assessment of the reconstituted chromatin fibers was performed via an electrophoretic band shift assay (0.7% agarose gel in 0.2 × TB). The titration with the largest band shift (and hence the highest saturation of the DNA template with histone octamers) was selected for single-molecule experiments.

**Flow cell preparation**. A clean coverslip was coated with 0.1% nitrocellulose (Ladd Research) in amylacetate solution and subsequently mounted onto a poly-dimethylsiloxane (Dow Corning) flow cell with a pre-cut channel. The flow cell was incubated with 10 ng/μl anti-digoxigenin (Sigma–Aldrich) for 2 h and then passivated with 4% bovine serum albumin (w/v) and 0.1% Tween-20 overnight at 4°C. Polystyrene reference beads (1 μm, Polysciences) were flushed into the channel and incubated at room temperature for 30 min. Then 20 ng/ml of reconstituted fibers in measurement buffer (100 mM KCl, 2 mM MgCl$_2$, 10 mM HEPES, pH 7.5, 10 mM NaN$_3$, 0.2% BSA, 0.1% Tween-20) was flushed into the flow cell and incubated at room temperature for 10 min. Magnetic beads from stock (either 1 μm diameter MyOne or 2.8 μm diameter M270 streptavidin-coated superparamagnetic beads, Invitrogen) were diluted 1000 times in the measurement buffer and injected into the flow chamber. Finally, untethered beads were flushed out with the measurement buffer. In the experiments with MyOne beads, the measurement buffer enriched with 250 ng/μl heparin was flushed into the flow cell after the force and twist measurements, in order to dissociate histone octamers from the tethered DNA.

**Magnetic tweezers**. The home-built magnetic tweezers setup used in this study was described previously[72]. After mounting the flow cell[71] on the microscope, the sample was illuminated from the top by a LED light (Lumitronix). The transmitted light was collected by a 60× oil-immersion objective (Olympus) and focused onto a CMOS camera (Teledyne Dalsa) operating at an acquisition rate of 58 Hz. The bead position was tracked with nanometer accuracy using a custom-written tracker software[73]. Forces were calibrated before by equating the power spectral density (PSD) of the bead motion to thermal energy[72]. Moving a pair of cubic magnets (1 mm gap) towards the tethered bead increased the force whereas rotating the magnets changed the degree of supercoiling of the tether (ΔLk) by ±1 per turn. The applied rotation rate was 1 turn per second. The bead position was tracked in real-time without filtering or averaging. Tethers were initially stretched up to 6 pN and directly refolded to the initial state. Subsequently, the rotational constraint of the tethers was checked. Because each DNA strand is attached to the flow cell surface and the magnetic bead via multiple bonds, the application of torque results in the change of its linking number. However, some tethers carry one or more nicks within the DNA backbone, which are introduced during purification and/or manipulation of the DNA. These tethers are not rotationally constrained, resulting in dissipation of the applied torque. To identify such rotationally unconstrained DNA, the chromatin tethers were stretched with 1.5 pN of force and subsequently twisted in the positive and negative direction (∼15 turns). The extension of a rotationally unconstrained (nicked) molecule remains constant throughout the entire twisting cycle. In contrast, a rotationally constrained molecule forms plectonemes under positive turns, which is detected by a decrease in its extension. When more than one DNA molecule is attached to a magnetic bead, the extension of the tether decreases abruptly, regardless of the directionality of rotation. After discarding the rotationally unconstrained and incorrectly tethered molecules, twist measurements were performed at different force regimes. Finally, force-extension curves were measured between 0 and 50 pN. The acquired data was analyzed using LabVIEW software by fitting the statistical mechanics model presented earlier[40,71]. Extension-twist curves were analyzed with the torsional spring model (explained in Supplementary Note 3).

**Magnetic torque tweezers**. To perform magnetic torque measurements, the pair of cubic magnets was replaced by a cylindrical magnet with a small side magnet attached on it (Supplementary Fig. 5), as shown in Kriegel et al.[52]. The chromatin fibers were stretched at ∼2 pN while 10 negative turns were applied. Forces were calibrated for individual tethers as in Yu et al.[72]. Upon magnet rotation, the bead traced out a circular trajectory, whose center was determined in order to assess the exact attachment point of the DNA to the bead. Using this information, the bead's X, Y coordinates could be translated into radial coordinates to determine subsequently the angular position of the tether[52,60,62,74]. Furthermore, the fluctuations of the relaxed tether were measured to quantify the angular trap stiffness. Finally, a series of measurements was performed in which various degrees of positive or negative twist was applied, while the corresponding shift in the angular position of

the bead was measured. The restoring torque present in the tethered molecule could be calculated at each value of the applied turns by multiplying the angular shift by the trap stiffness. The main criterium used to select torque curves for further analysis was the measured trap stiffness. When the tether is torsionally very stiff (above 300 pN·nm·rad$^{-1}$), typical torques exerted by the chromatin tether lead to very small angular shifts that are difficult to resolve. In contrast, if the trap stiffness is too low (below 30 pN·nm·rad$^{-1}$), the bead cannot reliably follow the rotation of the magnet, and the twist is not applied to the tether in a controlable manner[52]. The selected torque curves were offset such that the minimal values of the torques measured equaled the value of the DNA melting torque (-10 pN·nm)[60].

**Reporting summary**. Further information on research design is available in the Nature Research Reporting Summary linked to this article.

## Data availability

The data that support the findings of this study are available from the authors upon reasonable request. Source data underlying the reported averages in Figs. 1, 2 and Supplementary Figs. 2, 6 are provided as a Source Data file.

## Code availability

A C++/CUDA bead tracking algorithm integrated with the LabView 2011 measurement software is available at http://www.github.com/jcnossen/BeadTracker. Datasets presented in Fig. 2, Supplementary Figs. 2–4 were collected and analyzed with a custom measurement software written entirely in LabView 2014, which is available upon request. All remaining datasets were analyzed with custom-written MATLAB-2014b scripts, which are available upon request.

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

## Acknowledgements

We acknowledge Chi Pham, Theo van Laar and Ineke de Boer for the preparation of the DNA constructs and Gwendolyn Blonk for the help with experiments. We are grateful to Thomas B. Brouwer, Nicolaas Hermans, Richard Janissen and Kaley McCluskey for discussions and the support with the magnetic tweezers instruments. We also thank Mohamed Ghoneim for critical reading of the manuscript. This work was supported by the Netherlands Organisation for Scientific Research (NWO/OCW), as part of the Frontiers of Nanoscience program, by NWO-VICI Research Program Project 680-47-616, and by the European Research Council via Consolidator Grant DynGenome (312221).

## Author contributions

A.K and H.M. performed the experiments. H.M. and J.v.N. conceived the torsional spring model. A.K. and H.M. implemented the model to analyze the data. O.O. validated the torque measurements. J.v.N. and N.H.D. coordinated the study, and A.K., J.v.N., and N.H.D. wrote the paper.

## Competing interests

The authors declare no competing interests.
