## [Peer Review File · Nature Communications]

Reviewers' comments:

Reviewer #1 (Remarks to the Author):

In this manuscript by the Dekker and van Noort laboratories, the authors utilize sophisticated magnetic tweezers measurements in combination with elegant modeling to dissect in detail how reconstituted chromatin fibers respond to applied force and torque. Importantly, this study was carried out under near-physiological conditions that allow for the formation of compact fibers and nucleosome stacking. As such, this study presents a timely advance and fills an important gap in our mechanistic understanding of chromatin structure and its response to constraints imposed by chromatin-interacting machinery. Moreover, the data are of excellent quality, the experiments are well-executed and controlled for, and the manuscript is well written. As such, I am very happy to recommend the manuscript, essentially as is, for publication in Nature Communications, once the following minor points have been addressed.

1) Interestingly, for the 197 repeat (Fig. 4C) the numbers of nucleosomes and tetrasomes add up ($21+4=25$, the number of 601 repeats in this construct), but they do not for the 167 repeat (30 repeats according to the methods section, but the model has 26 nucleosomes and 9 tetrasomes, which yields 35 in total). The authors should clarify or explain this apparent discrepancy. Also, the authors only mention tetrasomes in the discussion section but never explicitly explain where they come from. It would increase accessibility of the manuscript if the authors could expand on this earlier in the text.

2) The authors should discuss how an array of highly "stable" 601 sequences might compare to a chromatin fiber of more "normal" sequence composition. On a different note, all of the 601 sequences are oriented in the same way. It would be interesting to know whether the 601 asymmetry would favor specific fiber configurations.

3) The elastic response of chromatin highlighted in this study suggests an important role of chromatin fibers in the transient buffering of twist. Even at the level of a single nucleosome, the transient buffering of single twist defects has recently been directly observed (Sabantsev et al. Nature Communications 2019). The authors could consider discussing this observation.

Reviewer #2 (Remarks to the Author):

In the manuscript titled "Chromatin fibers stabilize nucleosomes under torsional stress", the authors investigated the effect of twist on chromatin fibers using single molecule magnetic tweezers. Due to the double helical structure of DNA, torsion play an important role in DNA related process such as DNA replication and gene transcription. To date, it is still elusive how torsion regulate chromatin structure. In this study, the authors aimed to quantitatively analyze the effect of torsion on chromatin fiber using magnetic tweezers, which has potential interests in the field. My major concerns are described as follows.

(1)The assembled chromatin fiber used in the magnetic tweezers measurements should be analyzed by EM firstly, including negatively staining or cryo-EM. In this study, the authors assembled two different samples on tandem repeats of 601 DNA sequences (30×167 bp and 25×197 bp) and claimed that the two samples corresponded to solenoid and zig-zag chromatin fiber structure respectively. However, up to now, there is no direct and strong structural evidence for solenoid structure in vitro. Because the structure is critical to understand how twist regulates the twist number and writhe of DNA chain in the chromatin fiber, the authors should analyzed and confirm their sample structure carefully.

(2)The sample should be well controlled for uniform structure. The authors need monitor the reconstituted chromatin fiber by analytic ultracentrifugation and EM analysis. The authors carried out Force-Extension curve measurements and fitted the data with the theoretical torsional spring

model. Many data reveal that the samples are consisted of nucleosomes and tetrasomes indeed. The mixture of nucleosomes and tetrasomes cannot maintain a stable chromatin fiber structure and introduce uncontrollable factors in the measurements. Although chromatin presents various structure motifs in vivo, the sample assembled on tandem repeats of DNA sequences can be organized into uniform structure in vitro.

(3)The torsional spring model should be interpreted in more details. In this manuscript, there is no any description about the model. It's necessary for the authors to explain and confirm the theoretical model, which is important for the readers to better understand this model.

(4)About the long handles of the sample. As described in the Methods, the handles were ligated to the 601 DNA repeats. Then the histone octamer was mixed and reconstituted into chromatin fiber using salt dialysis. First, how to control or block nucleosome being assembled on the DNA handles. Second, in the major of the measurement including in the low force region and high force region, the supercoil happens on the two DNA handles, which cannot help to trace the structure dynamics of chromatin fiber. The authors should assemble the chromatin fiber with very short DNA handles and focus on the dynamics of chromatin fiber in the presence of torsion.

(5)The major conclusion of chromatin fiber folded into a left-handed superhelix is not fully supported by their data. Based on the measurements, the results reveal that chromatin fiber is twist buffer. When twist introduce into the twist-constrained chromatin fiber, two possible changes for DNA structure. One is the change of DNA twist number and the other is the change of DNA writhe. Both the changes are depended on the structure of chromatin fiber. Based on the measurements in this manuscript, it is difficult to make the conclusion that chromatin fiber folded into a left-handed superhelix.

The minor points are:

(1)There are some grammatical errors in the manuscript such as "kpbs" in line 99 and "per 1 helical turn" in line 43.

(2)The folding and unfolding dynamics of chromatin fiber has been studied in detail previously (Li et al., 2016, Molecular Cell 64, 120), they demonstrated that chromatin fiber containing 24x 177-bp or 187-bp tandem repeats of 601 DNA sequences shows two-start zig-zag chromatin fiber with tetranucleosomal unit, which is supported by their cryo-EM structures of reconstituted chromatin fiber in vivo (Song et al., Science, 2014). The authors showed that 197-bp chromatin fiber folded into a left-handed superhelix, and the authors need to discuss this discrepancy.

(3)In line 62, the authors claimed that "the linking number of a single nucleosome equals -1", how to make this conclusion?

(4)The measure buffer used in the experiments includes NaN₃, BSA and Tween20. The authors need to carry out some control experiments to test effects of these reagents on chromatin fiber structure.

(5)The length of DNA handle in line 99 is 2 kbp but in line 242 is 1.9 kbp, please clarify.

(6)The parameters used in the spring model such as persistence length and twist persistence length should be measured in the measured buffer.

(7)In the experiments for heparin, how to introduce the heparin into the flow cell and fix the linking number of tether.

Reviewer #3 (Remarks to the Author):

In their manuscript, Kaczmarczyk and coauthors describe a single-molecule, magnetic tweezers study of chromatin fibres, addressing the higher-order structure of chromatin and its mechanical (and in particular torsional) properties. Overall, this is a nice study, combining different magnetic tweezers tools in combination with modelling that provides important new insights in the structure of chromatin. In my opinion, it could be published in Nature Communications after the authors have addressed the following points.

- Overall writing. The manuscript, although generally well written, could be made a bit more

accessible to the broad audience of Nature Comm, which also consists of non specialists. For example: explain a bit more what tetrasomes are and why this is relevant for the current study. Some of the methods are also not described in much detail, for example (line 152/153): how is the torsionally (un)constrained chromatin tether obtained? How can the authors see the difference.

- The model. The authors should make a bit clearer in the abstract that they are using a model that has been published previously (and not a new one for this particular study). In addition, linking to the previous comment, some explanation of the model in the main text would be good. What is modelled, what are input parameters (and where do they come from), what is the output, what are variables. Most is now in small detail in the supplement, fine. But some higher-level explanation in main text might be good.
- Line 138-143. The text is a bit confusing about the numbers of the slope. In figure 1C it says 1.35, in the text 1.4 (for this specific tether), 1.35 on average. Please be a bit more clear.
- Line 223/224. "Restabilize nucleosome stacking and the reassembly of chromatin higher-order structure". This suggests that a structure is first broken and at even higher twist, the same structure is reestablished. Is this true? Or might another structure with interacting nucleosomes be formed at higher twist?
- Small textual things: a few times (e.g. l 115, 394, 398) an article (a / the) is missing. Discussion contains many sentences starting with "Here".
- I miss a discussion of what these studies of the artificial 601-sequence, with constant spacing say about naturally occurring nucleosomes: bound less tightly to the DNA, with different spacing. It would be good if the authors could discuss how their findings might connect to the in vivo situation.

Rebuttal to the manuscript “Chromatin fibers stabilized nucleosomes under torsional stress”

We thank the three Reviewers for their positive and constructive comments on our manuscript. We have carefully considered these and our responses are included in point-by-point fashion below (in blue). We believe that the resulting improvements to the manuscript address all of the remarks posed by the Reviewers, and hope that with this constructive review process our manuscript will be considered ready for publication in Nature Communications.

Kind regards,

The authors.

Response to Reviewer 1

“In this manuscript by the Dekker and van Noort laboratories, the authors utilize sophisticated magnetic tweezers measurements in combination with elegant modeling to dissect in detail how reconstituted chromatin fibers respond to applied force and torque. Importantly, this study was carried out under near-physiological conditions that allow for the formation of compact fibers and nucleosome stacking. As such, this study presents a timely advance and fills an important gap in our mechanistic understanding of chromatin structure and its response to constraints imposed by chromatin-interacting machinery. Moreover, the data are of excellent quality, the experiments are well-executed and controlled for, and the manuscript is well written. As such, I am very happy to recommend the manuscript, essentially as is, for publication in Nature Communications, once the following minor points have been addressed.”

We thank the Reviewer for his/her appreciation of our work.

1. Interestingly, for the 197 repeat (Fig. 4C) the numbers of nucleosomes and tetrasomes add up ($21+4=25$, the number of 601 repeats in this construct), but they do not for the 167 repeat (30 repeats according to the methods section, but the model has 26 nucleosomes and 9 tetrasomes, which yields 35 in total). The authors should clarify or explain this apparent discrepancy. Also, the authors only mention tetrasomes in the discussion section but never explicitly explain where they come from. It would increase accessibility of the manuscript if the authors could expand on this earlier in the text.

Number. Indeed, we observe some variation in the composition of individual chromatin fibers. The number of assembled nucleosomes depends mainly on the DNA:histone ratio used during the salt dialysis chromatin reconstitution. For this reason, we titrated a range of 601-DNA:histone stoichiometries and selected the samples with 1:1.3 ratio (*i.e.* above the stoichiometric 1:1 ratio), which showed best folding. For convenience of the magnetic tweezers experiments, we complemented the 601-array with relatively long handles to minimize protein-surface interactions (further explained in the response to Reviewer 2, point 4). We expect the excess histone complexes to be loaded in random fashion on this flanking DNA, which acts as an internal competitor for histones. As a result, the average sum of nucleosomes and tetrasomes over all studied molecules exceeded the number of 601 sequences (see histograms and captions of Fig. 2B and Fig. S2B).

Tetrasome composition. Nucleosomes that are not protected by neighboring nucleosomes (*e.g.* via interactions mediated by the histone tails) will have an increased tendency to partially disassemble into tetrasomes. This will hold particularly for the more weakly bound non-601 nucleosomes located on the DNA handles. To a reduced extent, this will also apply to nucleosomes at the peripheries of our chromatin fibers. Thus, we can explain both the existence of tetrasomes and why the number of fully folded nucleosomes is slightly lower than the number of 601 elements (25 out of 30 in Fig. 3C, and 21 out of 25 in Fig. 4C).

Consequences for chromatin fiber properties. In our earlier work (Kaczmarczyk *et al.* 2017, JBC), we reported quantitatively similar values for the stiffness and stacking energy of chromatin fibers including 15 repeats of the 601 sequences, as opposed to the 25 or 30 repeats used in this work. This shows that, provided that nucleosome-nucleosome interactions are maintained, the exact number of nucleosomes does not affect the overall elastic properties of the chromatin fiber.

To clarify the above points, we have expanded the information in our manuscript on tetrasomes:

* We now include a definition of tetrasomes (line 74-75).

* We now also report the number of nucleosomes and tetrasomes in the studied chromatin fiber in the Results (line 200). There, we refer to the discrepancy pointed out by the Reviewer: “For example, by fitting the force-

extension curve of the 167-NRL fiber in Fig. 3A, we inferred that it contained more than 30 assembled particles: 25 fully folded nucleosomes, and 9 tetrasomes that result from partial disassembly of H2A-H2B dimers (e.g. from excessive nucleosomes formed on the non-601 DNA handles). As evidenced by the presence of the unstacking transition at ~5 pN, the chromatin fiber remained fully folded below this force, allowing us to probe its response to turns.”

* We have added a paragraph (lines 367-381) discussing the effect of tetrasomes on our chromatin tethers.

2. The authors should discuss how an array of highly “stable” 601 sequences might compare to a chromatin fiber of more “normal” sequence composition. On a different note, all of the 601 sequences are oriented in the same way. It would be interesting to know whether the 601 asymmetry would favor specific fiber configurations. We have primarily used highly “stable” 601-sequences repeats because their usage ensures that nucleosomes are regularly spaced. By such control of nucleosome positioning, one can obtain chromatin fibers stabilized by histone-tail interactions. A number of recent studies of chromatin *in situ* (referred in the Introduction) has shown that groups of nucleosomes can form more complex structures that could be similar to uniform fibers.

While we agree that investigating the influence of 601-sequence orientation on chromatin fiber structure would be of interest, we believe that such a study would go beyond the scope of the present manuscript.

To address the Reviewer’s suggestion, we now phrase some of our findings in the context of native chromatin: * In the Introduction (line 93), we add: “By employing DNA with multiple repeats of 601-sequences, we obtained highly regular chromatin fibers. Previous work on isolated fragments of native chromatin showed remarkably similar unfolding behavior; however, large heterogeneities were observed in terms of composition (Hermans *et al.*, 2017, Sci Rep). Thus, the *in vitro* reconstituted fibers provide us with a platform to quantify the effects of mechanical stress in much more detail.”

* In the Discussion (line 360), we emphasize that the reported torsional properties are conserved between one-start and two-start 601-fibers. From this, we propose that other fiber-like secondary structures could similarly act as topological buffers.

* In the Discussion (line 482), we add a text describing “normal”, *in vivo* sequence composition: “Although nucleosome mapping studies showed that nucleosomes can be evenly distributed *in vivo* (Jiang *et al.*, 2009, Nat Rev. Genet; Baldi *et al.*, 2018, NSMB), they are likely to be less regular than the 601-arrays used in our study. Regular spacing may be reinforced by a variety of factors such as linker histones or chromatin remodelers (Happel *et al.*, 2009, Gene; Beshnova *et al.*, 2014, PLoS Comp Bio). These epigenetic regulators can potentially control the formation of condensed structural units that act as twist buffers, when necessary. DNA sequence is another important factor in nucleosome positioning and may play a role in higher-order folding (Eslami-Mossallam *et al.*, 2016, Adv in Col and Int Sci). The synthetic 601-sequence was used to impose controlled positioning, but how it affects the mechanical properties of folded fibers requires further study. As argued here, and in previous studies, 167-NRL and 197-NRL fibers fold into different higher-order structures. Nevertheless, both were shown to have a high capacity to absorb twist, suggesting that this may be a more universal feature of folded chromatin fibers.”

3. The elastic response of chromatin highlighted in this study suggests an important role of chromatin fibers in the transient buffering of twist. Even at the level of a single nucleosome, the transient buffering of single twist defects has recently been directly observed (Sabantsev *et al.*, Nature Communications 2019). The authors could consider discussing this observation.

We thank the Reviewer for suggesting this article (published during the preparation of our manuscript). Indeed, the observed discontinuity in the propagation of torsional stress along the DNA observed by Sabantsev *et al.* indicates that individual nucleosomes can buffer twist defects. That said, the reported phenomenon occurs at the scale of 1-3 bp, or ~2% of the nucleosomal DNA.

Because the main conclusion of Sabantsev *et al.* supports the previously reported plasticity of nucleosomes (Bancaud *et al.*, 2006, NSMB), we have added a reference to it (line 77), which reads: “Similar transient states were not detected in nucleosomes; yet, isolated nucleosomes can absorb some of the imposed torsional stress by buffering the twist defects in the DNA helix and thereby reduce the build-up of torque (Vlijm *et al.*, 2015, Cell Reports; Sabantsev *et al.*, 2019, Nat Comm).”

Response to Reviewer 2

In the manuscript titled “Chromatin fibers stabilize nucleosomes under torsional stress”, the authors investigated the effect of twist on chromatin fibers using single molecule magnetic tweezers. Due to the double helical structure of DNA, torsion play an important role in DNA related process such as DNA replication and gene transcription. To date, it is still elusive how torsion regulate chromatin structure. In this study, the authors aimed to quantitatively analyze the effect of torsion on chromatin fiber using magnetic tweezers, which has potential interests in the field

We thank the Reviewer for pointing out importance of our work and potential interest to the community.

1. The assembled chromatin fiber used in the magnetic tweezers measurements should be analyzed by EM firstly, including negatively staining or cryo-EM. In this study, the authors assembled two different samples on tandem repeats of 601 DNA sequences (30*167 bp and 25*197 bp) and claimed that the two samples corresponded to solenoid and zig-zag chromatin fiber structure respectively. However, up to now, there is no direct and strong structural evidence for solenoid structure in vitro. Because the structure is critical to understand how twist regulates the twist number and writhe of DNA chain in the chromatin fiber, the authors should analyzed and confirm their sample structure carefully.

The Reviewer suggests to perform EM imaging prior to magnetic tweezers investigations, in particular, to verify the secondary structure of chromatin fibers. We note several points:

- While it would be indeed highly desirable to support our findings by structural data that unequivocally resolve the topology of our fibers, we deem such measurements rather unrealistic. Despite many attempts in the community to acquire such data, only very few successful reports have been published (all are referenced in our manuscript). This is probably because of the highly flexible nature of the fibers, which results in rather disordered structures that are not suitable for crystallography or single-particle EM analysis. Regardless of why so few structural reports on chromatin fiber structure have been published, it is not likely that such approaches will be successful in resolving the path of the linker DNA in our fibers.
- Our chromatin fibers were prepared using the DNA plasmids and buffer conditions described in Robinson *et al.* (PNAS, 2006), including native gel electrophoresis to verify fiber quality. Robinson and co-workers extensively studied with EM very similar chromatin fibers but they could not directly resolve fiber topology, due to the limitations described above. They did infer, however, based on the relation between Nucleosome Repeat Length (NRL) and fiber diameter, that 197-NRL fibers folded into 1-start solenoid structures, consistent with our interpretation. Though microscopes have improved since and we are still in close contact with these authors, sample preparation of fibers, and the observed diversity of structures is still impeding high-resolution analysis.
- Our previous force-spectroscopy studies (Kruithof *et al.*, 2009, NSMB; Meng *et al.*, 2014, NAR, Kaczmarczyk *et al.*, 2017, JBC; de Jong *et al.*, 2018, BioPhysJ) all support the interpretation that 167-NRL fibers fold into a 2-start helix and 197-NRL fibers fold into a 1-start helix. Arguments included experiments on cross-linking of stacked fibers, increased stiffness and cooperativity of unstacking in 167-NRL fibers, Mg²⁺-dependence of folding and were supported by rigid base-pair Monte Carlo simulations of these structures. In the current study, the fibers reproduced previously reported behavior.
- The new experimental evidence presented here did not reveal major differences in response to torque between 167-NRL and 197-NRL fibers. This indicates that the fiber geometry may not be critical to how applying turns affects twist/writhe of the chromatin fiber, and this observation is not critically dependent on the precise topology.
- Our manuscript focuses on the mechanical response to force and torque of nucleosomal arrays with the emphasis on their unique elastic properties under physiological conditions that favor nucleosome stacking. The fact that nucleosomes stack and form secondary structures has been shown by X-ray crystallography, electron microscopy, computational simulations, and single-molecule force spectroscopy. While these studies have examined the chirality and flexibility of tetrasomes and unfolded arrays of nucleosomes, none have investigated the torsional properties of folded chromatin fibers. We, therefore, argue that our results are a significant new addition to the field and warrant publication by themselves.

2. The sample should be well controlled for uniform structure. The authors need monitor the reconstituted chromatin fiber by analytic ultracentrifugation and EM analysis. The authors carried out Force-Extension curve measurements and fitted the data with the theoretical torsional spring model. Many data reveal that the

samples are consisted of nucleosomes and tetrasomes indeed. The mixture of nucleosomes and tetrasomes cannot maintain a stable chromatin fiber structure and introduce uncontrollable factors in the measurements. Although chromatin presents various structure motifs in vivo, the sample assembled on tandem repeats of DNA sequences can be organized into uniform structure in vitro.

Here, the Reviewer suggests to perform EM imaging as a form of a 'sample control'.

In our view, single-molecule force spectroscopy is a sensitive and reliable assay to assess sample uniformity and resolve subtle variations between molecules (as presented quantitatively in Fig. 2B and Fig. S2B). The number of wrapped histone octamers can be counted unambiguously from the 25 nm unfolding steps in our force-extension curves (Fig. 2A, Fig. S2A). Given this sensitivity, and the variations that we observe, we can assess fiber integrity for each molecule separately. Although EM can resolve individual nucleosomes, it is impossible to characterize the same individual fiber with both techniques. Moreover, the high sensitivity and the ability to resolve differences between fibers cannot be matched with analytical ultra-centrifugation, or any other bulk technique, though we did carefully characterize each batch of fibers with native gel electrophoresis.

There are two factors that complicate the analysis of the chromatin fibers by complementary techniques:

- The addition of DNA handles allows for additional nucleosomes that do not constitute a folded chromatin fiber (see response to Reviewer 1).
- Introduction into the flow cell leads to dilution to extremely low concentrations and exposure to transient drag forces when the flow cell is flushed.

Both factors are potentially detrimental to the chromatin fiber, and any quality check therefore needs to be done after assembly of the glass-fiber-bead tether. Analysis before or after the measurement is either impossible or does not take these manipulations into account.

Regarding the presence of tetrasomes and how they affect the mechanical properties of the fiber, please see the response to Reviewer 1 (point 1). We agree that tetrasomes within 601 repeats could be deleterious to maintaining stable chromatin fiber structure.

To clarify this issue, we have emphasized in the revised manuscript that, prior to data analysis, we set a maximal number of tetrasomes tolerated in chromatin fibers (line 379):

"Either way, by discarding the molecules in which the number of fully folded nucleosomes was less than 90% of the number of 601-repeats, we ensured that the chromatin fiber was the dominant structure in the analyzed tethers."

3. The torsional spring model should be interpreted in more details. In this manuscript, there is no any description about the model. It's necessary for the authors to explain and confirm the theoretical model, which is important for the readers to better understand this model.

Due to length constraints, we explain only the torsional spring model's key concepts in the main text. There, we note that the model describes the response of two torsional springs (corresponding to a rigid DNA and a soft chromatin fiber), and that it is incorporated into the framework of our previously published statistical mechanics framework which quantitatively captures chromatin fiber unfolding (Meng *et al.*, 2015, NAR). Details of the model are explained in the Supplementary Information.

To test the proper functioning of the model, we analyzed the rotation-extension curves in Fig. S6B with its analytical solutions described in the Supplementary Information. In doing so, we obtained a value for the torsional persistence length of the DNA that is consistent with previously reported values (Lipfert *et al.*, 2010, Nat Met).

To further clarify these points, we have made the following changes in the manuscript:

* We have rephrased the description of the model and specified its main goal, namely the quantification of the chromatin twist modulus (lines 252-260).

* Subsequently, we have listed the fixed input parameters and variables of the model (lines 261 -268).

* We now state more clearly that proper functioning of the model was confirmed by fitting the bare DNA rotation- and torque curves (line 269-280).

4. About the long handles of the sample. As described in the Methods, the handles were ligated to the 601 DNA repeats. Then the histone octamer was mixed and reconstituted into chromatin fiber using salt dialysis. First, how to control or block nucleosome being assembled on the DNA handles. Second, in the major of the measurement including in the low force region and high force region, the supercoil happens on the two DNA

handles, which cannot help to trace the structure dynamics of chromatin fiber. The authors should assemble the chromatin fiber with very short DNA handles and focus on the dynamics of chromatin fiber in the presence of torsion.

Concerning the point about nucleosome formation on the DNA handles, please see our response to Reviewer 1 (point 1).

Regarding the DNA handles, its length (2030 bp corresponding to $\sim 0.6 \mu\text{m}$) was critical for several reasons:

- The dimension of a compact chromatin fiber with ~ 30 nucleosomes is only $\sim 30 \text{ nm} \times 60 \text{ nm}$, which is 20x fold less than the bead diameter ($1 \mu\text{m}$). In magnetic tweezers, the bead is not free to rotate in all directions. As a result, the attachment point of the DNA handle is fixed at any site of the bottom hemisphere of the bead. Moreover, the equatorial DNA-bead attachment is particularly preferred for tracking angular fluctuations in magnetic torque tweezers. Therefore, longer DNA handles increase the fraction of tethers in which the bead is sufficiently far away from the surface. Given the relatively low throughput of the technique, the longer chromatin tethers serve an important purpose.
- It is very beneficial to keep the nucleosomes away from the surfaces of the bead and the glass. Histones are known to be adhesive and any histone-surface contact prohibits quantitative interpretation of the experiments.
- Supercoiling of the DNA handles is an indicator of a tether's rotational constraint. This allows to easily discard the tethers with one or more nicks in the DNA backbone, as they are not responsive to torsion.
- Buckling torque of the DNA handles is a well-distinguishable feature and is one of the key parameters in the torsional spring model used to calculate the twist modulus per nucleosome. Capturing the buckling transition would be impossible with DNA handles that are shorter than a single plectoneme.

These are all technical points that facilitate the experiments, but do not affect the interpretation of our data; even with short handles, we would need to take into account its flexibility for quantitative interpretation. Since the torsional behavior of bare DNA has been well-characterized, we choose to benefit from the advantages of longer handles and describe extensively their implications in the manuscript.

5. The major conclusion of chromatin fiber folded into a left-handed superhelix is not fully supported by their data. Based on the measurements, the results reveal that chromatin fiber is twist buffer. When twist introduce into the twist-constrained chromatin fiber, two possible changes for DNA structure. One is the change of DNA twist number and the other is the change of DNA writhe. Both the changes are depended on the structure of chromatin fiber. Based on the measurements in this manuscript, it is difficult to make the conclusion that chromatin fiber folded into a left-handed superhelix.

The left-handedness of compact fibers was previously suggested based on the analysis of the EM images (Finch & Klug, 1976, PNAS; Robinson *et al.*, 2006, PNAS; Song *et al.*, 2014, Science). Our findings support this via three separate observations:

- The experimentally deduced linking number per nucleosome (which takes into account the twist and the writhe of the linker DNA) has a larger negative value than the linking number of a nucleosome in isolation, corresponding to a negative linking number for the looped linker DNA (Fig. S1).
- Similarly, the computationally deduced linking number change induced by twist-induced chromatin unfolding also has a negative value. By analyzing the rotation-extension curves at 5 pN (Fig. 3B, 4B, top traces) we quantified how many turns are required to unstack a chromatin fiber and deduced the overall twist/writhe of the linker DNA and the "outer" nucleosomal turn.
- The stability of the chromatin fiber is reduced upon the application of positive turns (which can be seen from the lower force required to induce chromatin unstacking, see Fig. S3A, S4A). If a chromatin fiber was right-handed (regardless of whether the underlying structure was zig-zag or a solenoidal helix), positive turns would over-twist it rather than destabilize it.

We have made the following textual changes:

* We now specify that our conclusions are in agreement with reports of the chromatin fiber chirality obtained by EM (lines 413-414).

* We have rephrased the sentence: "We demonstrated that both fibers adopt a left-handed chirality" to "We inferred that both fibers likely adopt a left-handed chirality" to indicate that the conclusion comes from indirect observations (lines 353-354).

The minor points:

(1) There are some grammatical errors in the manuscript such as “kpbs” in line 99 and “per 1 helical turn” in line 43.

We thank the Reviewer for pointing this out and have corrected these.

(2) The folding and unfolding dynamics of chromatin fiber has been studied in detail previously (Li *et al.*, 2016, Molecular Cell 64, 120), they demonstrated that chromatin fiber containing 24x 177-bp or 187-bp tandem repeats of 601 DNA sequences shows two-start zig-zag chromatin fiber with tetranucleosomal unit, which is supported by their cryo-EM structures of reconstituted chromatin fiber *in vivo* (Song *et al.*, Science, 2014). The authors showed that 197-bp chromatin fiber folded into a left-handed superhelix, and the authors need to discuss this discrepancy.

Song *et al.* indeed showed a two-start zig-zag geometry of chromatin fibers assembled on 177-bp and 187-bp tandem 601-repeats. Unfortunately, they did not report EM structures of chromatin fibers reconstituted on 197-NRL that we showed before to fold into a solenoid, one-start helix (see discussion above and the manuscript text). Note, however, that the referred structures were studied under very low salt conditions (hence the high-force and non-equilibrium unstacking transitions in the work of Li *et al.*), in contrast to a buffer used in our study that more closely reflects the physiological ionic strength.

We added the following paragraph (lines 317-325) to better reference this work:

“We note that recent cryo-EM studies have demonstrated zig-zag folding for fibers with intermediate NRL of 177-bp or 187-bp (Song *et al.*, 2014, Science), which were corroborated by force spectroscopy experiments by Li *et al.* (Mol Cell, 2016). For 197-NRL fiber, however, a high-resolution structure has not been so far reported. Earlier electron microscopy on chromatin fibers inferred an interdigitated solenoidal structure for fibers with such NRL, based on the relationship between NRL and fiber diameter (Robinson *et al.*, 2006, PNAS). Our force-spectroscopy data obtained on 197-NRL chromatin are most compatible with this interpretation. We note, that chromatin structure critically depends on other factors *e.g.* specific buffer conditions or the presence of linker histones (H1/H5) (Happel & Doenecke, 2009, Gene; Allahverdi *et al.*, 2015, Sci Rep), which could induce alternative fiber topologies.”

(3) In line 62, the authors claimed that “the linking number of a single nucleosome equals -1”, how to make this conclusion?

We added a line in the introduction (line 63) saying that this number was obtained experimentally on a DNA minicircle system established by Prunell *et al.* As explained in the introduction, this is consistent with the X-ray crystal structure of the nucleosome core (Luger *et al.*, Nature, 1997) that shows a writhe \$Wr = -1.7\$ per nucleosome and under-twisted nucleosomal DNA, resulting in a linker number of -1.

(4) The measure buffer used in the experiments includes Na₃N, BSA and Tween20. The authors need to carry out some control experiments to test effects of these reagents on chromatin fiber structure.

BSA and Tween-20 were essential to prevent the non-specific interactions of the sample with the flow cell surface. BSA is widely used in many surface-based experimental assays as a passivation and crowding agent that is neutral to biological samples. Its absence in the buffer would highly decrease the yield of our experiments. Moreover, one could argue that its presence better mimics the conditions found *in vivo*.

Sodium azide (NaN₃) served in our buffer as an antibacterial preservative. Force-extension curves recorded in a buffer without NaN₃ are not different from those obtained in a buffer containing the azide salt, provided that the removal of NaN₃ would be compensated with Na⁺ ions (in the form of NaCl) to maintain the same ionic strength of the buffer. In addition, we feel that it is important to keep the buffer conditions the same as in our previous work (Meng *et al.*, 2014, NAR; Kaczmarczyk *et al.*, 2017, JBC).

In our study, rather than exploring the effects of all buffer components on chromatin structure (which would indeed be very interesting, but warrants a separate study), we explicitly describe the buffer contents and follow other experimental approaches that use identical or similar conditions.

(5) The length of DNA handle in line 99 is 2 kbp but in line 242 is 1.9 kbp, please clarify.

We have now specified the correct, exact length of the DNA handles, which is 2030 bp. In some paragraphs, we approximate this length to ~2 kbp.

(6) The parameters used in the spring model such as persistence length and twist persistence length should be measured in the measured buffer.

Indeed, in the measurement buffer, we have validated the persistence length and twist persistence length by a set of force- and torque-measurements on bare DNA. Fig. S6A shows a representative force-extension curve of a bare DNA which, upon fitting to the Worm-Like-Chain model, yielded a value for the persistence length of 46 nm. Fig. S6C shows an averaged torque-turns curve which, upon linear fitting, yielded a value for the twist persistence length of 79 nm. Both numbers are consistent with previously reported values (Lipfert *et al.*, 2011, Nat Comm) and in agreement with the values obtained from fitting the rotation-extension curves (Fig. S6B).

(7) In the experiments for heparin, how to introduce the heparin into the flow cell and fix the linking number of tether.

The magnet above the flow cell keeps the linking number of the tether constant, as the beads cannot rotate. By injection of the measurement buffer with heparin, histones dissociate, but the linking number of the DNA does not change. The linking number of the tether can only change upon rotation or removal of the magnetic field, but this is maintained constant here.

To clarify this procedure, we have expanded the information concerning heparin introduction in line 141: "Dissociation of a single histone octamer from rotationally constrained DNA increases the contour length by > 50 nm but does not change the linking number of the tether, unless the magnetic field is removed or rotated."

Response to Reviewer 3:

In their manuscript, Kaczmarczyk and coauthors describe a single-molecule, magnetic tweezers study of chromatin fibres, addressing the higher-order structure of chromatin and its mechanical (and in particular torsional) properties. Overall, this is a nice study, combining different magnetic tweezers tools in combination with modelling that provides important new insights in the structure of chromatin. In my opinion, it could be published in Nature Communications after the authors have addressed the following points.

We thank the Reviewer for this positive feedback.

1. Overall writing. The manuscript, although generally well written, could be made a bit more accessible to the broad audience of Nature Comm, which also consists of non specialists. For example: explain a bit more what tetrasomes are and why this is relevant for the current study. Some of the methods are also not described in much detail, for example (line 152/153): how is the torsionally (un)constrained chromatin tether obtained? How can the authors see the difference.

We have addressed similar comments on tetrasomes in our response to Reviewer 1 (point 1).

To clarify how we distinguish between torsionally (un)constrained tethers:

* We add a short note (line 164) explaining the origin of the torsionally unconstrained molecules: "these tethers were selected to have nicks in one of the DNA strands, see Methods."

* In the Methods (line 542-552), we describe in more detail how to recognize rotationally unconstrained (nicked) tethers: "However, some tethers carry one or more nicks within the DNA backbone, which are possibly introduced during purification and/or manipulation of the DNA. These tethers are not rotationally constrained, resulting in dissipation of the applied torque. To identify such rotationally unconstrained DNA, the tethers were stretched with 1.5 pN of force and subsequently twisted in the positive and negative direction (~15 turns). The extension of a rotationally unconstrained (nicked) molecule remains constant throughout the entire twisting cycle. In contrast, a rotationally constrained molecule forms plectonemes under positive turns which is detected by its decreasing extension. When more than one DNA molecule is attached to a magnetic bead, the extension of the tether decreases abruptly, regardless of the directionality of rotation. After discarding the rotationally unconstrained and incorrectly tethered molecules, twist measurements were performed at different force regimes. Finally, force-extension curves were measured between 0 and 50 pN."

2. The model. The authors should make a bit clearer in the abstract that they are using a model that has been published previously (and not a new one for this particular study). In addition, linking to the previous comment, some explanation of the model in the main text would be good. What is modelled, what are input parameters (and where do they come from), what is the output, what are variables. Most is now in small detail in the supplement, fine. But some higher-level explanation in main text might be good.

Please see our response to Reviewer 2 (point 3).

3. Line 138-143. The text is a bit confusing about the numbers of the slope. In figure 1C it says 1.35, in the text 1.4 (for this specific tether), 1.35 on average. Please be a bit more clear.

To clarify the discrepancy, we have added the information in the caption of Figure 1C that the value of the population-averaged linking number is -1.35 (corresponding to the result of the linear fit in the inset). The linking number of the individual tether shown in the main panel equals -1.4.

4. Line 223/224. "Restabilize nucleosome stacking and the reassembly of chromatin higher-order structure". This suggests that a structure is first broken and at even higher twist, the same structure is reestablished. Is this true? Or might another structure with interacting nucleosomes be formed at higher twist?

The Reviewer's second statement is correct. Indeed, as indicated in the main text, the maximal extension of the tethered chromatin fiber shown in the rotation-extension curve (panel B of Fig. 3 and 4) corresponds to the extension of the same molecule at 7 pN (panel A of Fig. 3 and 4), which is the force regime at which nucleosome-nucleosome interactions are broken and the nucleosomal DNA is partially unwrapped from the nucleosomes. Further application of turns causes a decrease of the extension, indicating the refolding of the chromatin fiber. We expect the newly formed fiber to have opposite handedness.

To emphasize our novel observation and its interpretation, we now explicitly mention it in the Results (line 237): "Thus, the unstacking transition of chromatin is facilitated by the application of positive turns, but excessive positive turns restabilize nucleosome stacking and the reassembly of chromatin higher-order structure, most likely with an opposite handedness."

- Small textual things: a few times (e.g. l 115, 394, 398) an article (a / the) is missing. Discussion contains many sentences starting with "Here".

In general statements, we do not think it is necessary to include articles. However, we made the following edits:

* Added: "a" to line 123

* Added: "the" to line 442

* In the two sentences in the Discussion, the word "here" was changed to other phrases such as: "in this paper" (line 346) and "moreover" (line 353).

I miss a discussion of what these studies of the artificial 601-sequence, with constant spacing say about naturally occurring nucleosomes: bound less tightly to the DNA, with different spacing. It would be good if the authors could discuss how their findings might connect to the in vivo situation.

Please see our response to Reviewer 1 (point 2).

Reviewers' comments:

Reviewer #1 (Remarks to the Author):

The authors have carefully and rigorously addressed all of my concerns and I am therefore happy to recommend publication of this manuscript in Nature Communications.

Reviewer #2 (Remarks to the Author):

The authors have improved the manuscript and have clarified some questions we raised. There are still a few questions needed to be clarified:

(1)The major concern is still about the chromatin sample used in this study. The well-reconstituted chromatin fiber in vitro is the key issue for the magnetic tweezers experiments in this study.

Although the authors claimed that the chromatin fibers were well prepared, the authors can carry out AFM experiments to visualize the chromatin fibers reconstituted in vitro. In addition, the state of nucleosomes assembled on the template with the long DNA handles had better to be examined experimentally before the single molecule experiments.

(2)The second concern is about the theoretical model. As the authors mentioned, they measured the persistence length (46 nm) and twist persistence length (79 nm). However, the authors fitted with persistence length 50 nm and twist persistence length 100 nm. The author need to fit the experimental data with these measured parameters under their buffer conditions.

(3)The third concern is the magnetic tweezers experiments. The measurements were carried out in the buffer with NaN3, BAS and Tween20. The author claimed that these reagents were essential to prevent the non-specific interactions. We suggest that the flow cell is incubated in the NaN3, BAS and Tween20 buffer for an adequate time before measurements, then the stretching experiments for chromatin fibers are carried out in a relative clean buffer.

(4) As far "the linking number of a single nucleosome equals -1", the proper description would be "the change of linking number of a single nucleosome equals -1".

Reviewer #3 (Remarks to the Author):

The authors answered my comments thoroughly and have improved the text accordingly. I fully support publication of the manuscript in its current version in Nature Communications.

Rebuttal to the manuscript “Chromatin fibers stabilized nucleosomes under torsional stress”

We thank the Reviewers for their positive comments on our revised manuscript and the support for publication. Below, we have addressed the remaining comments by Reviewer 2 as fully as we can.

We hope that with the accompanying modifications our manuscript will be considered suitable for publication in *Nature Communications*.

Kind regards,
The authors.

Response to Reviewer 1

“The authors have carefully and rigorously addressed all of my concerns and I am therefore happy to recommend publication of this manuscript in *Nature Communications*.”

We thank the Reviewer for his/her recommendation to publish our manuscript.

Response to Reviewer 2

“The authors have improved the manuscript and have clarified some questions we raised. There are still a few questions needed to be clarified:

(1) The major concern is still about the chromatin sample used in this study. The well-reconstituted chromatin fiber in vitro is the key issue for the magnetic tweezers experiments in this study. Although the authors claimed that the chromatin fibers were well prepared, the authors can carry out AFM experiments to visualize the chromatin fibers reconstituted in vitro. In addition, the state of nucleosomes assembled on the template with the long DNA handles had better to be examined experimentally before the single molecule experiments.”

In Reviewer 2’s initial comments, he/she requested that we perform additional measurements (EM, analytic ultracentrifugation) to verify our sample quality. We have extensively explained in our previous Rebuttal why this is neither realistic nor informative (see pp. 3-4 of our previous rebuttal). In short, we are of the opinion that the sample interrogation as performed in the flow cell of the magnetic tweezers provides the most quantitative and thorough investigation of chromatin fiber that can be performed at present.

We regret that Reviewer 2’s additional suggestion to perform AFM studies would similarly not be sufficiently informative. As an example of the image quality that we achieved using such an approach in the past, we present Fig. R1 at right. While one can clearly distinguish the nucleosome-free handles, assessing fiber structure or the exact state of the nucleosomes (stacked or not, partially unwrapped or not) is not feasible. Furthermore, one would have to carry out an entire series of controls to ensure that the mica surface onto which the molecules are directly deposited would not disrupt chromatin fiber structure. We are not only of the opinion that this is beyond the scope of this study, but furthermore that it would not serve to either complement or enhance the exact and quantitative information that we have deduced from our magnetic tweezers study.

*Figure R1. Nucleosomes assembled onto a DNA including an array of 601 sequences (15*601) flanked by non-601 DNA. Imaging is performed on a mica surface in a buffer with 100 mM KCl and 10 mM HEPES, pH 8.0.*

Regarding the “state of nucleosomes”, we can only interpret this request as a concern about possible defects of nucleosomes, i.e. missing H2A-H2B dimers (resulting in tetrasomes). We explained in our previous Rebuttal how we were able to distinguish tetrasomes from nucleosomes in the magnetic tweezers and discussed the implications of their presence. Again, as one can appreciate from Fig. R1, this information cannot be obtained from AFM images.

(2) The second concern is about the theoretical model. As the authors mentioned, they measured the persistence length (46 nm) and twist persistence length (79 nm). However, the authors fitted with persistence

length 50 nm and twist persistence length 100 nm. The author need to fit the experimental data with these measured parameters under their buffer conditions.

We indeed measured the persistence length of 46 nm of the specific bare DNA molecule presented in Figure S6A. The number was obtained by fitting the data to the Worm-Like-Chain model (Bouchiat et al., Biophys J 76:409, 1999), as is standard in the field. We analysed many more molecules with this approach, and obtained values for the persistence length that range between 40 and 55 nm. Therefore, in the torsional spring model, we deliberately fixed the value for the persistence length to 50 nm, which is the typical literature value for similar salt conditions. Any variation on the scale of a few nanometers is in any case too small to be of influence on the outcome of the torsional spring model.

Regarding the twist persistence length: in the main text of our manuscript, we explicitly distinguish the effective twist persistence length (C_t), which is force-dependent, and the twist persistence length (C), which is force-independent. The experimentally obtained value of 79 nm is representative of the former (C_t) as it was measured at specific tension (~ 2 pN). In the model, we calculated the effective twist persistence length using the formula shown in Supplementary Information (line 36) that incorporates the force-independent twist persistence length as a constant ($C = 100$ nm, conform established literature values).

** For clarity, in the main text, we have now modified the symbol of the force-independent twist persistence length (now C_{lim} , previously C), to remain consistent with the abbreviation used in the Supplementary Information.

(3) The third concern is the magnetic tweezers experiments. The measurements were carried out in the buffer with NaN₃, BSA and Tween20. The author claimed that these reagents were essential to prevent the non-specific interactions. We suggest that the flow cell is incubated in the NaN₃, BSA and Tween20 buffer for an adequate time before measurements, then the stretching experiments for chromatin fibers are carried out in a relative clean buffer.

We are very confident that these components are neutral to our samples at low concentration. They have been standard in the field of single-molecule experiments with DNA-protein interactions for decades and are not associated with any nefarious interactions with nucleosomes or any other proteins. Leaving out BSA and Tween-20 from the buffer in which the experiments are performed is undesirable as it results in aspecific sticking of the chromatin fibers to the flow cell surface (irrespective of preceding passivation with the same components). Use of low concentrations of NaN₃ is standard to inhibit buffer contamination by microorganisms and attendant nucleases, which would otherwise be detrimental to DNA molecules, in particular torsionally constrained ones.

(4) As far “the linking number of a single nucleosome equals -1”, the proper description would be “the change of linking number of a single nucleosome equals -1”.

We thank the Reviewer for this suggestion. We have corrected the sentence in the main text.

Response to Reviewer 3

“The authors answered my comments thoroughly and have improved the text accordingly. I fully support publication of the manuscript in its current version in Nature Communications.”

We thank the Reviewer for his/her recommendation to publish our manuscript.